# Recognition of Wood-Boring Insect Creeping Signals Based on Residual Denoising Vision Network

**DOI:** 10.3390/s25196176

**Published:** 2025-10-05

**Authors:** Henglong Lin, Huajie Xue, Jingru Gong, Cong Huang, Xi Qiao, Liping Yin, Yiqi Huang

**Affiliations:** 1College of Mechanical Engineering, Guangxi University, Nanning 530004, China; 18770625991@163.com; 2Shenzhen Branch, Guangdong Laboratory for Lingnan Modern Agriculture, Genome Analysis Laboratory of the Ministry of Agriculture and Rural Affairs, Agricultural Genomics Institute at Shenzhen, Chinese Academy of Agricultural Sciences, Shenzhen 518120, China; huangcong@caas.cn (C.H.); qiaoxi@caas.cn (X.Q.); 3Technical Centre for Animal, Plant, and Food Inspection and Quarantine, Shanghai Customs, Shanghai 200002, China; llgod@126.com (H.X.); kalen_17@163.com (J.G.); yinliping@hotmail.com (L.Y.)

**Keywords:** residual denoising, feature extraction, comparison experiments, recognition system

## Abstract

Currently, the customs inspection of wood-boring pests in timber still primarily relies on manual visual inspection, which involves observing insect holes on the timber surface and splitting the timber for confirmation. However, this method has significant drawbacks such as long detection time, high labor cost, and accuracy relying on human experience, making it difficult to meet the practical needs of efficient and intelligent customs quarantine. To address this issue, this paper develops a rapid identification system based on the peristaltic signals of wood-boring pests through the PyQt framework. The system employs a deep learning model with multi-attention mechanisms, namely the Residual Denoising Vision Network (RDVNet). Firstly, a LabVIEW-based hardware–software system is used to collect pest peristaltic signals in an environment free of vibration interference. Subsequently, the original signals are clipped, converted to audio format, and mixed with external noise. Then signal features are extracted through three cepstral feature extraction methods Mel-Frequency Cepstral Coefficients (MFCC), Power-Normalized Cepstral Coefficients (PNCC), and RelAtive SpecTrAl-Perceptual Linear Prediction (RASTA-PLP) and input into the model. In the experimental stage, this paper compares the denoising module of RDVNet (de-RDVNet) with four classic denoising models under five noise intensity conditions. Finally, it evaluates the performance of RDVNet and four other noise reduction classification models in classification tasks. The results show that PNCC has the most comprehensive feature extraction capability. When PNCC is used as the model input, de-RDVNet achieves an average peak signal-to-noise ratio (PSNR) of 29.8 and a Structural Similarity Index Measure (SSIM) of 0.820 in denoising experiments, both being the best among the comparative models. In classification experiments, RDVNet has an average F1 score of 0.878 and an accuracy of 92.8%, demonstrating the most excellent performance. Overall, the application of this system in customs timber quarantine can effectively improve detection efficiency and reduce labor costs and has significant practical value and promotion prospects.

## 1. Introduction

With the continuous advancement of global trade, the volume of international timber circulation has significantly increased, leading to a growing risk of the cross-border transmission of wood-boring pests. Such pests are highly prone to spreading with cargo flow during timber transportation. By boring into the internal structure of timber, they not only severely reduce the economic value of timber but also cause the invasion of alien harmful species, posing a potential threat to agricultural and forestry ecosystems. Currently, customs mainly rely on manual visual inspection for timber pest detection. Professional personnel observe signs of insect holes on the timber surface and make judgments by combining means such as splitting timber. These wood-boring insects are in their infancy and are weak and difficult to detect, as shown in Figure 1. However, this method has problems such as long detection time, high labor costs, low efficiency, and being susceptible to subjective experience, making it difficult to meet the needs of rapid quarantine for large-scale and high-throughput import and export timber. Therefore, achieving the rapid, accurate, and intelligent detection of wood-boring pests inside timber has become a research hotspot in the field of pest quarantine [1]. In recent years, the academic community has carried out extensive explorations on detection technologies for wood-boring pests, particularly focusing on the identification of weak vibration signals generated by their feeding or peristalsis. The research path has roughly gone through three stages: (1) detection by analyzing the pulse interval range corresponding to signal characteristic spectrograms or signal-based algorithms; (2) detection by combining signal characteristic spectrograms with machine learning methods; and (3) detection by combining signal characteristic spectrograms with deep learning methods.

In the field of wood-boring pest detection, early studies predominantly employed methods based on signal characteristic spectrogram analysis and traditional signal processing algorithms, achieving the preliminary identification of pest infestations by extracting features from weak vibration signals generated by insect activities. For instance, Guo et al. utilized pickups and recorders to collect crawling sound signals of stored grain pests (*Tribolium confusum* and *Oryzaephilus surinamensis*) and digitally processed the collected signals via Matlab software [2,3]. Their approach involved low-pass filtering to eliminate background noise, followed by extracting acoustic functional spectrum features of crawling sounds and identifying pest species by comparing energy value differences in power spectrum pulses among different species. Geng et al.used a microphone to collect adult crawling sound signals of *Tribolium castaneum* and *Sitophilus oryzae* [4,5]. After low-pass filtering and wavelet threshold denoising, he analyzed the main frequency and sub-frequency characteristics in wheat medium. The study revealed that although the main frequencies of power spectra for different pests were close, the sub-frequency distributions exhibited significant differences, serving as effective distinguishing criteria. In foreign studies, Mankin et al. conducted mean spectral energy distribution analysis on crawling and feeding sounds of larvae such as *Rhynchophorus ferrugineus*, *Monochamus alternatus*, and *Buprestidae*, primarily using characteristic parameters such as high-frequency pulse energy, pulse burst frequency, and pulse interval for species differentiation [6]. Dingfeng Lou collected feeding sounds of six pest species through a microphone in an anechoic chamber, plotted their power spectral density diagrams, and determined the main energy distribution frequency bands for species identification [7]. At the signal algorithm level, Mingzhen Zhang recorded crawling and turning vibration signals of *Sitophilus zeamais* and *Tribolium castaneum* on a film in a soundproof room [8]. He used low-pass filtering and wavelet thresholding for denoising, and combined with the FastICA algorithm to achieve signal centering, whitening, and independent component extraction, thereby separating mixed signals of different pest species. Shenghuang Liu, for the vibration signals of three species of longhorn beetle larvae, firstly applied Variational Mode Decomposition (VMD) for denoising, then extracted the energy proportion of each node through three-layer wavelet packet decomposition, and achieved species identification by integrating the fluctuation duration of time-domain signals with the main frequency components of the frequency domain [9]. The above studies indicate that in relatively quiet or ideal experimental environments, methods based on spectrograms and traditional signal processing algorithms exhibit certain effectiveness in identifying wood-boring pests. However, in practical applications, due to the complex and changeable environmental noise, the above methods have limited ability to suppress real background noise, often leading to a decline in recognition accuracy or even failure. Therefore, such methods still have significant limitations in terms of noise robustness, environmental adaptability, and generalization ability in real-world scenarios, necessitating further optimization and improvement.

Subsequently, studies introducing machine learning methods based on feature spectrograms for pest identification gradually attracted attention in the academic community. Some scholars attempted to combine pest activity signal features with statistical learning models to enhance the automation and accuracy of identification. For example, Min Guo collected crawling and turning sound signals of two stored grain pests, *Sitophilus zeamais* and *Tribolium castaneum* in a soundproof environment [10]. She firstly used Mel-frequency cepstral coefficients (MFCCs) to extract frequency-domain feature parameters, then estimated the parameters through a Gaussian Mixture Model (GMM) combined with the Expectation–Maximization (EM) algorithm, and used a clustering algorithm to classify and identify pest sound signals. Mingzhen Zhang, for four types of activity sound signals of two stored grain pests in a soundproof box, used the Isometric Feature Mapping (ISOMAP) method for manifold dimensionality reduction to extract streamline features in pest sound signals [11]. Subsequently, a Support Vector Machine (SVM) with a heavy-tailed radial basis function as the kernel was used to construct a classification model, which trained and tested streamline features data to achieve the effective differentiation of pest species. Yufei Bu, from the perspective of time and frequency domains, extracted features such as the pulse duration and energy distribution range of seven pest species, and used the sum of squared deviations for the cluster analysis of time-domain data to achieve pest species identification and classification [12]. Additionally, Ping Han proposed an automatic parameter selection method for Support Vector Machines (SVMs) based on chaotic optimization, aiming at the parameter selection problem of SVM models in the sound signal identification of stored grain pests [13]. This method guides the search process of parameters C and kernel width σ by generating chaotic sequences of logistic mapping and circular mapping and extends chaotic variables to the parameter space through “carrier mapping” to achieve the global optimization of SVM parameter combinations. Experimental results show that this method not only improves the recognition accuracy but also effectively reduces the number of models and improves the overall calculation efficiency. Although the above methods have achieved certain recognition effects in controlled environments, there are still some key problems in practical applications. Firstly, the assumptions about the distribution of pest signals in the modeling process of some methods deviate from the real data, leading to insufficient model generalization ability. Secondly, most algorithms are sensitive to hyperparameters, and improper parameter selection may significantly affect the classification performance. Especially when facing new pests not involved in training or complex environmental interferences, the recognition effect is prone to an obvious decline. Therefore, constructing an identification model that is robust to diverse pest species and has good adaptability to noise and feature disturbances remains a key challenge in current research.

In research on the integration of signal feature spectrograms and deep learning methods, a series of breakthroughs have been made in recent years. Some scholars have attempted to introduce neural network architectures to enhance the robustness and accuracy of pest sound signal identification under complex backgrounds. For example, Ping Han used the adaptive neural network noise reduction method (Madaline) for signal filtering and combined it with an adaptive silencer to effectively suppress environmental noise interference for three typical stored grain pests: *Sitophilus oryzae*, *Sitophilus zeamais*, and *Tribolium castaneum* [14]. The experiment set up two groups of control signals: one group was a mixture of pest sounds and noise, and the other was pure noise signals. In the study, pest spectrograms were input into a Backpropagation neural network (BP neural network) for training, and pest species classification and identification were achieved through labeled samples. Xiaoqian Tuo constructed a noise reduction neural network (Enhance) based on a dilated convolution structure for processing insect sound data under three types of noise conditions [15]. Subsequently, four recognition models (InsectFrames) with different output dimensions of convolutional layers were designed to evaluate the differences in recognition accuracy under different feature expression capabilities. Juhu Li collected four types of signals: *Agrilus planipennis*, *Cryptorhynchus lapathi*, their mixed sounds, and environmental noise [16]. He extracted cepstral coefficient spectrograms through the MFCC method and input them into the self-designed deep neural network model BoreNet for classification and identification. To improve the model’s generalization ability, noise-free insect sound fragments were used in the training stage to capture universal signal features, and noisy insect sound fragments were introduced in the test stage to simulate practical application scenarios. Weizheng Jiang used sensors to collect signals of the emerald ash borer, *Holcocerus insularis*, their mixed sounds, and noise and also extracted MFCC spectrograms as model inputs [17]. He proposed a novel convolutional neural network architecture called the Residual Mixed-domain Attention Module Network (RMAMNet), which integrates channel attention and temporal attention mechanisms to enhance the model’s ability to learn key features, demonstrating good recognition stability in multi-source insect sound mixed backgrounds. Haopeng Shi proposed a compact and excellent-performance vibration-enhanced neural network for the larvae of the wood-boring pest *Agrilus planipennis* [18]. The network combines frequency-domain enhancement and time-domain enhancement modules in a stacked framework. Experimental results show that the enhanced network significantly improves the recognition accuracy under noisy backgrounds, exhibiting obvious performance advantages compared to the undenoised model. Overall, the above studies demonstrate the potential of combining deep learning with signal feature spectrograms in the identification of wood-boring pests. However, these methods still have problems such as high model complexity, long inference time, and limited generalization ability. Especially under unstructured noise and complex on-site background conditions, they still face challenges such as accuracy degradation and insufficient robustness. Therefore, constructing an efficient insect sound detection model with lightweight structure, strong feature perception ability, and adaptability to complex environments remains the core problem to be solved in this field.

In terms of detection methods, they are mainly divided into acoustic signal-based detection methods and vibration-based detection methods. In terms of acoustic detection, Chunfeng Dou used an NI acquisition card combined with an acoustic emission sensor SR 150 N to collect acoustic signals [19] and then used wavelet packets to reconstruct the time–frequency-domain signals of pests. The effect of larval number on the number of pulses, duration, and amplitude of the signal was studied. Yufei Bu used the AED-2010L sound detector (built-in sound sensor) combined with the SP-1L probe to collect four types of acoustic signals of two types of longhorn larvae [20] and distinguished them by the amplitude, waveform, pulse and energy of the spectrogram of the time domain map. Senlin Geng used microphones and sound capture cards to collect the sounds of two types of grain storage pests in the soundproof room [5] and distinguished them by the power spectrum. In terms of vibration signal detection: Piotr Bilski uses a CCLD accelerometer and acquisition card to collect vibration signals and distinguish pest signals from background noise through a Support Vector Machine [21]. Xing Zhang used the SP-1L piezoelectric sensor probe combined with the self-developed vibration sensor to collect vibration signals and designed TrunkNet to identify pest vibration signals [22].

Although current methods based on signal feature spectrograms and deep learning have achieved high accuracy in pest sound signal identification, two prominent issues remain. On the one hand, existing denoising networks perform insufficiently in handling non-uniform complex noise, limiting noise reduction effects, and on the other hand, most methods decouple denoising and classification processes, lacking a unified integrated modeling framework, which affects the robustness and efficiency of the overall identification system. To address the above issues, this paper proposes an integrated denoising and classification multi-attention recognition network—the Residual Denoising Vision Network (RDVNet). The model firstly performs deep denoising processing on pest sound signals through two groups of residual structures (each composed of four residual blocks), then inputs the denoised results into a lightweight classification network with a sandwich structure to complete end-to-end recognition tasks. On the collected actual insect sound dataset, RDVNet achieved excellent performance in both noise reduction performance and classification accuracy, verifying the effectiveness and practicality of the model design. The main contributions of this paper are as follows: (1) We propose an integrated multi-attention recognition network, effectively fusing the denoising module and the classification module. It significantly improves the model’s adaptability to non-uniform noise environments and recognition accuracy and enhances the network’s perception of key signal features under complex backgrounds. (2) We introduce a dedicated denoising module into the recognition model and conduct comparative experiments combined with three types of cepstral coefficient diagrams (MFCC, RASTA-PLP, and PNCC), verifying that PNCC features have the best expression effect in noisy environments. RDVNet achieves the best denoising performance among all comparative models. (3) We develop a PyQt-based visual recognition system, realizing full-process integration from signal collection, preprocessing, and feature extraction to recognition result display. It significantly improves data processing efficiency and visual interaction performance and has good practicality and promotion potential.

## 2. Experiments and Methods

### 2.1. Data Acquisition and Processing

To ensure the accuracy and stability of pest creeping signal collection, the experimental system was built in a relatively quiet and low-vibration indoor environment. The timber samples used in the experiment were provided by the customs department. Firstly, all timber samples were preliminarily screened, and their surfaces were manually observed for the presence of insect holes. After screening out timber samples with obvious insect holes, they were collectively used for subsequent signal collection experiments. In the preparation process, an electric drill and an electric saw were used to expand the insect holes until the presence of insects was observed. Subsequently, two CT1500L piezoelectric acceleration sensors (Chengke Electronic Technology Co., Ltd., Shanghai, China) were arranged near the insects in the wormholes. One end of each sensor was fixed at the position of the insect hole in the depth of the timber, and the other end was connected to the channel ports of the NI four-channel signal acquisition card through transmission cables, respectively. The signal acquisition card was connected to the computer through a USB port to realize the acquisition, transmission, and control of sensor signals. The data acquisition system was implemented by the LabVIEW software platform [23], and its operation interface is shown in Figure 2. The system program includes four main functional modules: virtual channel creation, sampling parameter setting, time–frequency domain amplitude curve visualization, and real-time data recording and saving. In this Figure, a represents acceleration, and sampling represents how much data is collected per second. Sampling point and length per channel mean the number of sample points transferred from the acquisition card to the software cache each time. FFT and phase represent the frequency and phase changes after fast Fourier transform. The CT1500L sensor has a sensitivity of 5000 mV/g and a measurement range of ±1 g, with high micro-vibration response capability. A sampling rate of 32 kHz was chosen for the experiment [24]. The primary effective frequencies of vibration signals generated by wood pests (such as crawling and gnawing) are concentrated in the 0–10 kHz range [25]. According to the Nyquist sampling theorem, the sampling rate should be at least twice the highest frequency component of the signal. Therefore, choosing 32 kHz ensures that the high-frequency characteristics of insect vibrations are captured without distortion while also balancing storage and computational efficiency. Each sampling cycle included 3200 sampling points. The sampling length of each channel was synchronously set to 3200 points per time, which could effectively capture the characteristics of weak vibration signals generated by insect activities.

All data collection experiments were conducted in a relatively quiet, low-vibration indoor environment, away from large machinery and vehicle traffic to minimize external vibration sources. Anti-seismic facilities are installed at the bottom of the test bench, which attenuated low-frequency mechanical vibrations transmitted from the ground. A total of four CT1500L piezoelectric sensors were used in the experiment. Since the maximum distance at which the sensor can detect pest vibration signals is 200 mm, two sensor probes are placed in holes drilled at different positions within 200 mm from the pests to collect data. The other sensors were independently placed on the surface of table as a control group for environmental background noise. Firstly, we selected the timber sample to be tested, completed the equipment connection, and determined whether there are obvious insect hole features on its surface. If insect holes were detected, we carried out appropriate reaming treatment on the holes.Until the pest is found, holes are drilled at different locations within 200 mm from the pest and sensor probes are inserted to collect data. If no insects were found in the sample, we replaced it with other timber for repeated testing. The collected original pest signals were generated into time-domain waveform data by the LabVIEW system, followed by format conversion and slicing processing. Each segment of the signal is uniformly divided into 30 s audio clips. For each audio clip, we extracted its time-domain amplitude change curve and the frequency-domain amplitude spectrogram obtained by Short-Time Fourier Transform (STFT) to observe whether there are obvious pulse characteristics of pest activities. If characteristic pulse signals are detected in the time-frequency domain, the audio data segment is retained and included in the dataset as a sample. If no pulse signals are detected, the signal segment will be discarded. The actual collection process is shown in Figure 3.

During the data collection process, the LabVIEW main interface displays the signal information collected by sensors in real time, including the time-domain amplitude variation curve of the original signal, the frequency-domain amplitude spectrogram processed by the Fast Fourier Transform (FFT), and the corresponding phase spectrogram. The synchronous display of the three groups of images indicates that the system acquisition module has been successfully started and entered the normal operation state. To more realistically simulate the vibration interference in the actual detection environment, the experiment synchronously collected the same number of vibration noise signals. The main sources of noise are the operating sounds of the surrounding machines, the footsteps of people passing by, and the voices of people, as well as the loud noises from vehicles during their driving. Subsequently, the pest peristaltic signals and noise were mixed with five different signal-to-noise ratios (SNRs), specifically set as −10 dB, −7.5 dB, −5 dB, −2.5 dB, and 0 dB, wherein the SNR represents the ratio of clean insect sound to noise signal. A negative SNR indicates that the amplitude of pest activity signals is significantly lower than the background noise, simulating the detection scenario under weak signal conditions. Considering the limitations of traditional time-frequency domain images in describing signal detail features, this paper further converts all mixed audio signals into corresponding cepstral coefficient images, including three types of feature spectrograms, MFCC, PNCC, and RASTA-PLP, and uses them as inputs to the subsequent deep learning recognition model.

### 2.2. Cepstral Coefficient Spectrogram Extraction

MFCC, PNCC, and RASTA-PLP are typical speech design features, rather than end-to-end methods. In this study, they map the one-dimensional vibration sequence output by the accelerometer into a two-dimensional feature map, which is then input into the subsequent network. In the research on vibration signal identification of wood-boring pests, MFCC is currently one of the most widely used feature spectrogram extraction methods.The conversion process is as follows: Firstly, the audio is pre-emphasized, framed, and Hamming-windowed. Then, each frame of the windowed signal is subjected to an FFT to obtain the spectrum. The power spectrum energy is weighted onto each Mel filter to obtain the filter bank energy. The Mel filter bank energy is logarithmized, and the logarithmic energy sequence is subjected to a discrete cosine transform and liftering to obtain the MFCC cepstrum, as shown in Figure 4. The MFCC maps audio signals to the frequency-domain space of the Mel filter bank to extract more sensitive cepstral feature spectrograms, and the specific implementation process can be found in relevant studies [26]. However, the MFCC is relatively sensitive to environmental noise, especially non-stationary noise and sudden interference, and has limited robustness under complex backgrounds. To improve the anti-interference ability of the model in actual customs detection scenarios, this paper firstly introduces two more robust cepstral feature extraction methods: PNCC and RASTA-PLP. Among them, PNCC has strong suppression ability for non-stationary noise and transient interference, while RASTA-PLP performs excellently in suppressing steady-state background noise and channel distortion. Considering that the acoustic environment at customs sites often includes both steady-state background noise (such as the continuous operation sound of transportation equipment), non-stationary noise (such as operational impact sound), and sudden interference, the above two methods can more comprehensively extract effective features from pest signals.

The overall processing flow of the two methods is shown in Figure 5, both including the following steps: common signal preprocessing operations such as pre-emphasis, framing, windowing, and Short-Time Fourier Transform (STFT). The main difference between the two lies in that RASTA-PLP introduces a dual-filtering structure. The first step uses a Bark filter bank to construct a critical band power spectrum. The second step uses an IIR band-pass filter to model the dynamic characteristics of the log power spectrum [27], suppressing steady-state distortion and slowly changing noise backgrounds. The core transfer function of the RASTA filter is shown in Equation (Equation 1), where *z* represents the unit delay operator (Z-transform variable) in the complex domain, and H(z) is the transfer function of the filter. In addition, RASTA-PLP also includes two key links, equal loudness pre-emphasis and intensity loudness nonlinear mapping, which are used to simulate the perceptual sensitivity to different frequency bands (such as the 40 dB equal loudness curve) and the power law relationship between sound intensity and loudness perception, respectively. Finally, after obtaining the time-domain envelope through inverse Fourier transform, the Durbin algorithm is used for all-pole modeling of the power spectrum, constructing a 12th-order linear prediction analysis model, and calculating 16-dimensional cepstral coefficients for subsequent model training and classification identification [28].(1)$$H\left(z\right)=0.1\cdot \frac{2+z^{-1}-z^{-3}-2z^{-4}}{z^{-4}\left(1-0.98z^{-1}\right)}$$

PNCC is a robust signal feature extraction method, especially suitable for processing complex environments with non-stationary noise. The method employs a gammatone filter bank for initial filtering during the signal preprocessing stage to obtain the power spectral representation of the signal across frequency bands [29]. Subsequently, PNCC achieves feature extraction through a series of modeling processes of perceptual mechanisms. The main steps as follows: (1) Medium-time power analysis: Perform sliding time window averaging on each band power signal with a window length of 65.6 ms (equivalent to 5 frames) to smooth short-term fluctuations and suppress short-term unstructured noise. (2) Asymmetric noise suppression: Track the lower envelope of the power signal through an asymmetric filter with dual-parameter adjustment capability to estimate the minimum energy level of background noise. This estimated value is subtracted from the original power, and a half-wave rectification operation is used to truncate negative values to zero, retaining the effective signal [30]. (3) Temporal masking: This mechanism dynamically tracks the power peak of each frequency band and suppresses transient signals below a certain proportion of the peak, further enhancing the signal robustness and highlighting important structural peak features [31]. (4) Frequency smoothing and power normalization: After completing the above steps, perform sliding average smoothing on the remaining power spectrum within the adjacent frequency band range (±4 filter channels), and normalize it with the long-term average power (time constant approximately 4.6 s) to alleviate the impact of speech intensity fluctuations over time. (5) The use of 1/15th power compression transform: To avoid the problem of logarithmic functions over-amplifying low-power noise, PNCC uses 1/15th power law compression instead of the traditional logarithmic compression method, making the spectral intensity distribution after noise suppression more balanced and contributing to the subsequent improvement of feature stability. (6) Cepstral coefficient generation: Apply Discrete Cosine Transform (DCT) to the compressed spectrum, extract the first 13-dimensional static PNCC coefficients [32], and further combine them with first- and second-order delta dynamic coefficients to finally form a 39-dimensional feature vector as input for deep learning models [33]. Its core formula is shown in Equation (Equation 2), used to calculate the background noise estimate for the m-th frame and l-th frequency band channel, where $\tilde{Q}\left[m,l\right]$ represents the medium-time power envelope output by the asymmetric filter.(2)$${\tilde{Q}}_{out}\left[m,l\right]=\left\{\begin{array}{cc} \lambda_{a}{\tilde{Q}}_{out}\left[m-1,l\right]+\left(1-\lambda_{a}\right){\tilde{Q}}_{in}\left[m,l\right], & \mathrm{if}\;{\tilde{Q}}_{in}\geq {\tilde{Q}}_{out}\\ \lambda_{b}{\tilde{Q}}_{out}\left[m-1,l\right]+\left(1-\lambda_{b}\right){\tilde{Q}}_{in}\left[m,l\right], & \mathrm{otherwise} \end{array}\right.$$

### 2.3. Development of PyQt Software

During the test, four sensors can be arranged at equal distances on the wood to collect signals within each range, as shown in the Figure 6, and the system processes the data into feature maps. The acquisition time is 120 s. The system calls the feature maps of each sensor and the weight file to perform inference. Each sensor will output a classified image. The output images are saved in two folders, one is “insect” and the other is “no-insect”. The proportion of the two folders to the total number is counted. We compare the proportion of “insect” folders in each sensor with the set threshold. If the insect ratio of a sensor exceeds the threshold, the wood detected by the sensor is considered to have insects. Due to the good accuracy of the model and to account for external noise, the threshold is set to 50% here. The threshold can be adjusted dynamically based on the model’s recognition performance. Through this PyQt system, users can complete the entire process of data processing and model testing without manually operating command-line scripts, which greatly improves the system’s usability and experimental efficiency.

To improve data processing efficiency and the visual interactive experience of model inference, this paper develops a Graphical User Interface (GUI) system based on PyQt, realizing full-process visual operations from data preprocessing, feature extraction to model inference. The system not only enhances the intuitiveness and controllability of the experimental process but also has good usability and portability. It can generate executable software through packaging, facilitating deployment and distribution on different platforms. Figure 7 shows the modular organizational structure diagram of the system, and Figure 8 is a schematic diagram of the system’s main interface. The system’s interface is overall divided into three major functional areas: (1) Data Processing Module: Users firstly select .tdms files generated by LabVIEW from wood as the input source of pest signals and choose background noise files for mixing processing. The system verifies the validity of the selected path. If the path is invalid, it prompts the user to reselect. After correct selection, users click the “tdms-wav” button, which executes the conversion operation, converting .tdms files in-to .wav format audio and mixing them with noise files according to the set signal-to-noise ratio. Subsequently, users can sequentially click the “wav-mfcc”, “wav-pncc”, and “wav-plp” buttons to complete the batch generation of three types of cepstral feature spectrograms. (2) Model Input and Inference Module: The system allows users to select an image folder (containing multiple images) from the generated feature maps as the inference input while loading the pretrained weight file. After clicking the “Inference” button, the system automatically invokes the specified model to perform forward prediction and records the inference time and classification results. (3) Result Display and Saving Module: The inference results will be displayed in the middle of the interface in the form of “image + corresponding predicted category”. Users can save the result images and choose to exit the system after the task is completed. All output images and results will be automatically stored in the specified local path for subsequent analysis and reuse. In the Figure 7: classification result table represents each image obtained by prediction is described as follows: index, saving path, category, probability

## 3. Image Denoising-Based Vision Recognition Model

### 3.1. Dataset Partitioning

The pest vibration signals and their mixed noise signals collected in this study were respectively used to construct the denoising model dataset and the recognition model dataset, and then the performance of three types of cepstral feature spectrogram extraction methods was evaluated under various signal-to-noise ratio conditions, including MFCC, PNCC, and RASTA-PLP. In the experiment, five SNR levels were set, −10 dB, −7.5 dB, −5 dB, −2.5 dB, and 0 dB, to simulate detection scenarios with different intensities of noise interference in real environments. For each SNR, the above three feature extraction methods were used to generate corresponding cepstral spectrograms as inputs for subsequent model training and evaluation. In the denoising model experiment, two types of sample data were constructed: one type was pure insect vibration audio, and the other type was insect vibration audio mixed with external noise. In the recognition model experiment, two types of samples were also constructed: one type was insect vibration audio mixed with external noise, and the other type was pure noise audio. In each experiment, 5000 feature images were collected and generated for each type of sample.There are 10,000 images at each signal-to-noise ratio. The dataset division ratios used for all models are as follows: the ratio of training set to test set is set to 80%:20%, and the training set is further divided into a validation set with the ratio of training set: validation set = 80%:20%. To ensure the consistency of the three types of feature spectrograms during the comparison, the signal preprocessing parameters were uniformly set as follows: sampling rate—32,000 Hz, Short-Time Fourier Transform window length—1024 points, and frame step size (Hop Length)—512 points. The uniform parameter setting facilitates a fair comparison of the denoising effects and recognition performances of different methods under various SNR conditions.

### 3.2. Denoising Module

Although significant progress has been made in denoising research on vibration signals of wood-boring pests in recent years, most existing methods still have certain limitations. Most existing methods adopt a series structure, only acting on a single dimension of the time domain or frequency domain, resulting in a single information processing path, feature redundancy. It weakens interaction between each stage of feature extraction, limiting the expression ability and generalization performance of the overall model. To solve the above problems, this paper cites a denoising network with a dual-branch parallel structure, named de-RDVNet [34], as shown in Figure 9. It takes three types of cepstral coefficient spectrograms (MFCC, PNCC, and RASTA-PLP) as inputs, fully fusing the multi-dimensional feature information of pest signals. The model is specially designed for classification tasks and has the following structural advantages: (1) Multi-scale feature fusion capability: de-RDVNet includes two different types of modules. The upper branch adopts Residual Attention Blocks (RABs) to focus on local detail feature extraction, and the lower branch uses Hybrid Dilated Residual Attention Blocks (HDRABs) to capture multi-scale global structural information. The two have significant complementarity in local–global and multi-scale modeling, which can improve the robustness to different types of noise. (2) Enhanced attention mechanism: The RAB module introduces a Spatial Attention Mechanism (SAM), and the HDRAB module introduces a Channel Attention Mechanism (CAM). They respectively focus on important feature regions in the spatial and channel dimensions, achieving saliency region enhancement and redundant feature suppression and improving the discriminability of feature expression. (3) Global feature fusion and long skip connections: The network uses a cross-branch feature concatenation strategy for fusion while introducing long skip residual connections within sub-modules, effectively alleviating the gradient vanishing problem in deep networks, enhancing context information integration, and improving training stability. (4) Parallel computability: The dual-branch structure design naturally supports GPU parallel computing, which can shorten inference time and reduce redundant computational overhead while maintaining high feature capacity, balancing accuracy and efficiency.

As shown in Figure 9: The proposed denoising network de-RDVNet is mainly composed of two types of sub-modules, RAB and HDRAB. The entire network adopts a dual-branch parallel structure to model the feature information in insect signals. All convolution kernels in the network are set to a size of 3 × 3, and the number of channels in each layer of feature maps is uniformly set to 128. The overall input and output channels of the network are both 3 (RGB channels), while between the intermediate connection nodes and the RAB and HDRAB modules, a feature representation with 64 channels is used, aiming to compress intermediate features and improve computational efficiency. The entire feature extraction process is as follows: Firstly, the input cepstral coefficient spectrograms (MFCC, PNCC, or RASTA-PLP) are processed by the initial convolutional layer and then sent to the two parallel branches of RAB and HDRAB for feature extraction, respectively. In the first half of the RAB module, two downsampling operations are set, and the image size is reduced through a 2 × 2 convolution with stride (stride = 2) to expand the receptive field and generate low-resolution feature maps. Correspondingly, in the second half of the RAB module, two upsampling operations are used, specifically 2 × 2 transposed convolution (stride = 1, padding = 0), combined with tensor dimension transformation to achieve the reconstruction and restoration of feature maps. Through the symmetric structure of downsampling and upsampling, the model can effectively restore the spatial resolution of the original image while preserving semantic information. Between the RAB and HDRAB modules, the network designs a long skip connection mechanism, which directly connects shallow features to deep structures, effectively alleviating the gradient vanishing problem, enhancing the modeling ability of detail information, and improving the fusion effect between features. Finally, the feature maps from the RAB and HDRAB branches are subjected to residual cascade fusion at the end and fused with global feature information in the input stage. Feature integration and mapping are performed through a convolution block and residual connection structure to generate the final denoised output image.

As shown in Figure 10a,b, RAB is composed of a residual structure and a Spatial Attention Module (SAM). The residual structure includes four basic residual blocks, each composed of two standard convolutional layers (Conv) and a Rectified Linear Unit (ReLU) [35], and realizes cross-layer information fusion through residual connections to extract rich local spatial features. Each convolutional layer has a kernel size of 3 × 3 and 128 channels. The SAM module in RAB is used to model the saliency regions in the spatial dimension, and its structure consists of the following components: Global Max Pooling (GMP), Global Average Pooling (GAP), 1 × 1 convolution, ReLU activation function, and Sigmoid activation function. GAP and GMP extract spatial summary features of the entire image from different statistical perspectives. Then, feature fusion is performed through convolutional layers and nonlinear activation. Thereafter, the weight map output by the Sigmoid layer is element-wise multiplied with the feature output by the residual block to achieve feature weighting enhancement. Finally, the weighted feature output by SAM and the input initial cepstral feature maps are added through the residual path to form the output features of the module, thus achieving the unity of local feature extraction, important region enhancement, and global information fusion. HDRAB is specially designed to capture long-range dependencies and multi-scale contextual information and is composed of a hybrid dilated residual structure and a Channel Attention Mechanism (CAM). The hybrid dilated residual structure includes four sub-modules, and each sub-module is composed of two convolutional layers (s-DConv) with different dilation rates and a ReLU activation function. The dilation rate s ranges from 1 to 4 to construct convolutional layers with different receptive fields and perform cascaded residual learning. This structure can expand the receptive field without significantly increasing the number of parameters and effectively extract multi-scale contextual features. The structure of CAM consists of GAP, the convolutional layer, ReLU, and Sigmoid. GAP is used to extract channel statistical information, then generate attention weights in the channel dimension through nonlinear transformation, and perform channel-wise multiplication with input features to highlight feature channels with strong expression capabilities. Finally, the weighted features and original features are fused through residual connections to output feature maps with enhanced channel attention. The designs of RAB and HDRAB realize saliency region modeling from the spatial dimension and channel dimension, respectively, collaboratively constructing a denoising module with local–global information coupling capability and feature adaptive regulation capability, significantly improving the model’s robustness and expression efficiency in complex backgrounds.

### 3.3. Residual Denoising Vision Network

Traditional attention mechanisms have been widely applied to image classification tasks, especially demonstrating excellent feature modeling capabilities in Transformer-series models. However, existing Transformer models generally have the following problems: Firstly, each attention head in the Multi-Head Self Attention (MHSA) needs to process the entire input feature map [36], lacking an effective feature partitioning and division of labor mechanism, resulting in high computational complexity and large memory consumption. Secondly, when the original Transformer structure faces complex, low-signal-to-noise ratio scenarios such as wood-boring pest signal identification, it lacks built-in anti-noise capabilities and struggles to adapt to the fine-grained feature modeling requirements under high-noise backgrounds. To address this, this paper cites a classification module based on the aforementioned denoising sub-network de-RDVNet and proposes a Residual Denoising Vision Network (RDVNet) with anti-noise capability.

Its main features are reflected in the following three aspects: (1) Sandwich Layout: To enhance the information channel interaction capability of the attention mechanism, this paper proposes a new module construction method called Sandwich Layout, whose calculation process is shown in Equation (Equation 3). This structure embeds a single MHSA layer between two Feed-Forward Network (FFN) layers [37]. Compared with the original Transformer structure, it effectively reduces the memory overhead caused by tensor rearrangement and element-wise operations in MHSA. By increasing the number of FFN layers, it enhances the nonlinear interaction between different channels and improves the performance of shallow attention networks in spatial detail expression. This design not only simplifies structural complexity but also enhances the model’s ability to model heterogeneous features. (2) RDV Block (Explicit Decoupling Attention Calculation Structure): RDVNet introduces module-level partitioned RDV Blocks, which group input features and allocate them to multiple attention heads, with each head only responsible for processing a subset of input features, thus achieving explicit attention decomposition calculation. The specific operation is shown in Equations (4)–(6). Different to traditional MHSA where each head processes complete features, RDV Block has the following advantages: reducing the computational burden of each attention head and improving operational efficiency, enhancing global feature expression capability through cascaded connections of output features among multiple attention heads, introducing a cross-head information interaction mechanism, which can proactively remove redundant features and improve the effectiveness of attention representation and memory utilization. This structure is particularly suitable for wood-boring insect sound identification with abundant background interference information, effectively highlighting structural movement patterns. (3) Over-Parameter Redistribution Strategy: Aiming at the difference in parameter sensitivity among different modules in the attention mechanism, this paper proposes a parameter redistribution strategy to dynamically adjust the channel resource distribution of different sub-modules—expanding the channel width of key modules such as the value projection layer to enhance their ability to represent pest features in high-dimensional space. Meanwhile, compressing the hidden layer dimensions in the FFN reduces redundant parameters in insensitive modules. This strategy achieves the compression of model complexity while ensuring performance, not only improving the model’s feature representation capacity but also optimizing inference speed and memory efficiency [38].(3)$$X_{i+1}=\prod^{N}\Phi_{i}^{F}\left(\Phi_{i}^{A}\left(\prod^{N}\Phi_{i}^{F}\left(X_{i}\right)\right)\right)$$

Here, $X_{i}$ denotes the input features of the i-th block. $\Phi_{i}^{F}$ represents the i-th FFN layer. $\Phi_{i}^{A}$ corresponds to the i-th self-attention layer (MHSA). N indicates the stacking count of FFN layers (the default is 1).(4)$$\begin{array}{cc} {\tilde{X}}_{ij} & =\mathrm{Attn}\left(X_{ij}W_{ij}^{Q},\;X_{ij}W_{ij}^{K},\;X_{ij}W_{ij}^{V}\right) \end{array}$$(5)$$\begin{array}{cc} {\tilde{X}}_{i+1} & =\mathrm{Concat}{\left[{\tilde{X}}_{ij}\right]}_{j=1}^{h}W_{i}^{L} \end{array}$$(6)$$\begin{array}{cc} X_{ij}^{\prime } & =X_{ij}+{\tilde{X}}_{i(j-1)},\;1<j\leq h \end{array}$$
where $X_{\mathrm{ij}}$: the input feature segment of the j-th head. $W_{\mathrm{ij}}^{Q}$, $W_{\mathrm{ij}}^{K}$, and $W_{\mathrm{ij}}^{V}$: the projection matrices for query, key, and value. $W_{i}^{L}$ denotes the output linear projection matrix, and *h* is the total number of attention heads. Attn(·) denotes the self-attention computation $\left(\mathrm{Softmax}(QK^{T})V\right)$.

The classification module is shown in the lower half of Figure 11: After completing the denoising process, the proposed classification module receives the denoised image as input and first splits and embeds the image into a high-dimensional feature space through an overlapping image patch embedding module. Specifically, the module consists of three sets of convolutional units (Conv2d), with each convolutional layer followed by Batch Normalization and a ReLU activation function [39]. This structure reduces the spatial resolution of the image through layer-by-layer convolution operations while increasing the number of channels, thus completing the mapping of the image to high-dimensional feature embedding and laying the foundation for feature interaction in the subsequent attention mechanism. The embedded features will be sequentially input into three groups of trunk structures composed of cascaded attention modules and the FFN, with the specific structure shown in Figure 12. Each module is wrapped by two fully connected FFN layers and a multi-head attention module, forming an “FFN–Attention–FFN” sandwich structure to enhance the cross-channel modeling capability of features. Subsequently, the feature maps pass through two groups of downsampling modules in sequence, and each group uses a downsampling layer composed of Conv2d and BatchNorm to achieve further feature compression. The core of the recognition module is the Token Interaction mechanism, which includes the following four key steps: (1) Feature partitioning: Divide the feature map into multiple sub-blocks along the channel dimension, with each sub-block corresponding to an attention head. (2) Query–Key–Value Projection (QKV Projection): Generate Query vectors (Q), Key vectors (K), and Value vectors (V) for each sub-block respectively. (3) Attention calculation: Calculate the attention score between the query and key through dot product, and weight the value vector after softmax normalization to generate the attention output. (4) Feature fusion: The output features of all attention heads are concatenated and integrated into the final feature representation through a linear projection layer. Finally, the fused feature map is input into a Global Average Pooling layer and then sequentially passes through a batch normalization layer and a linear classification layer to complete the final classification. The calculation process is shown in Equation (Equation 7), from which the prediction scores for each category can be obtained, and the category with the highest score is selected as the final recognition result output.(7)$$y=W\cdot BN\left(x\right)+b$$
y represents the classification score, and W is the weight matrix, while b is the bias. BN(x) normalizes the input features as shown in Equation (Equation 8). Firstly, calculate the mean and variance, and then perform standardization and affine transformation.(8)$$\mathrm{BN}\left(x\right)=\gamma \frac{x-\mu }{\sqrt{\sigma^{2}+\epsilon }}+\beta$$

### 3.4. Parameter Settings

For the proposed denoising sub-network module, a combined loss function is adopted during the training process to simultaneously optimize pixel accuracy and structure preservation capability. Specifically, the model loss consists of two components: the L1 Loss [40], which is used to measure the pixel-level absolute error between the model’s predicted image and the real clean image, defined as in Equation (Equation 9), and the Structural Similarity Index Measure (SSIM) loss, which is used to evaluate the consistency of images in terms of brightness, contrast, and structure, defined as in Equation (Equation 10). The final total loss function is a weighted combination of the two, as shown in Equation (Equation 11). This design aims to balance the detail restoration of images and the preservation of overall structural information, improving the subjective and objective quality of denoised images. To optimize network parameters, the AdamW optimizer is used during training, which combines the advantages of the traditional Adam optimizer and Weight Decay. Different to conventional weight decay, AdamW separates the weight decay term from the gradient update process, enabling the model to have a more stable regularization effect during parameter updates, improving convergence speed and generalization ability. Additionally, to improve training efficiency and convergence performance, this paper introduces a cosine annealing learning rate scheduler, which dynamically adjusts the learning rate in a cosine function manner, allowing it to gradually decrease during training. Therefore, it maintain stable model convergence in the later training stages, suppressing the risk of oscillation and overfitting.(9)$$L_{L1}=\frac{1}{N}\sum_{i=1}^{N}\left|y_{i}-{\hat{y}}_{i}\right|$$
where N is the total number of pixels, $y_{i}$ is the true value, and $\hat{y_{i}}$ is the predicted value.(10)$$SSIM\left(x,y\right)=\frac{\left(2\mu_{x}\mu_{y}+C_{1}\right)\left(2\sigma_{xy}+C_{2}\right)}{\left(\mu_{x}^{2}+\mu_{y}^{2}+C_{1}\right)\left(\sigma_{x}^{2}+\sigma_{y}^{2}+C_{2}\right)}$$

Here, $\mu_{x}$ and $\mu_{y}$ represent the means of images x and y, respectively, $\sigma_{x}^{2}$ and $\sigma_{y}^{2}$ denote the variances of images x and y, $\sigma_{\mathrm{xy}}$ is the covariance between images x and y, and $C_{1}$ and $C_{2}$ are stabilization constants. $\sigma_{\mathrm{xy}}$ is the covariance between unnoised images x and denoised y.(11)$$L_{total}=\alpha \cdot L_{L1}+\beta \cdot \left(1-SSIM\right)$$

Among them, α=0.8,β=0.2. The parameter update formula for AdamW is shown in Equation (Equation 12) below:(12)$$\begin{array}{cc} m_{t} & =\beta_{1}\cdot m_{t-1}+\left(1-\beta_{1}\right)\cdot g_{t}\\ v_{t} & =\beta_{2}\cdot v_{t-1}+\left(1-\beta_{2}\right)\cdot g_{t}^{2}\\ \theta_{t} & =\theta_{t-1}-\eta \cdot \frac{m_{t}}{\sqrt{v_{t}}+\epsilon }-\lambda \cdot \theta_{t-1} \end{array}$$

Here, $m_{t}$ and $v_{t}$ represent the first-order and second-order moment estimates, respectively, $g_{t}$ denotes the gradient, $\beta_{1}$ and $\beta_{2}$ are the decay coefficients, η is the learning rate, ε is the numerical stability parameter, and λ is the weight decay coefficient. The cosine annealing learning rate scheduler is shown in Equation (Equation 13):(13)$$\eta_{t}=\eta_{min}+\frac{1}{2}\left(\eta_{max}-\eta_{min}\right)\left(1+\cos\left(\frac{T_{cur}}{T_{max}}\pi \right)\right)$$

Here, $\eta_{t}$ denotes the current learning rate, $\eta_{\min}$ and $\eta_{\max}$ represent the lower and upper bounds of the learning rate, $T_{\mathrm{cur}}$ indicates the current iteration count, and $T_{\max}$ is the total number of iterations.

In the joint denoising and classification recognition task, the training of RDVNet uses the common cross-entropy loss function as the objective function to measure the difference between the model’s output class probability distribution and the true labels. The denoising module and the classification module jointly use cross entropy loss. The optimizer is consistent with the denoising module, and the AdamW optimizer is selected to fully combine the advantages of adaptive gradient updates and weight decay strategies, improving the model’s generalization ability and convergence stability. The training and testing of all models were completed on a hardware platform equipped with an NVIDIA Ge-Force RTX 4060 GPU to ensure that the experiments were carried out in a unified and reproducible experimental environment, achieving stable training performance and inference efficiency.

## 4. Results and Discussion

### 4.1. Denoising Comparison

In this study, the collected pest vibration signals were respectively converted into three types of cepstral coefficient spectrograms, MFCC, PNCC, and RASTA-PLP, which were used to characterize pure insect peristaltic signals and insect peristaltic signals mixed with noise. To simulate various complex noise backgrounds, datasets under five different signal-to-noise ratio (SNR) conditions were constructed [41], specifically −10 dB, −7.5 dB, −5 dB, −2.5 dB, and 0 dB, and proportionally divided into a training set, validation set, and test set. In the model evaluation experiment, the proposed denoising network de-RDVNet was systematically compared with multiple mainstream denoising models, including, VDNNet, RIDNet, CBDNet, and DeamNet. These networks are representative of signal denoising tasks and are often used as benchmarks. We used the same AdamW optimizer, cosine learning rate schedule, batch size, and loss function. We also used the same input feature map parameters (MFCC, RASTA-PLP, and PNCC). The comparison content included the differences in denoising performance of each model under the input of the above cepstral spectrograms and the quality of spectrogram feature restoration. To comprehensively measure the denoising effect of the model, this paper uses the following two common image quality evaluation indicators: Peak Signal-to-Noise Ratio (PSNR): Used to measure the difference between the restored image and the original image, it is usually used to evaluate the performance of image restoration tasks such as image compression and denoising. The larger the PSNR value, the closer the restored image is to the original image, and the better the noise suppression effect. Its calculation formula is shown in Equation (Equation 14). Structural Similarity Index Measure (SSIM): This is based on the joint modeling of brightness, contrast, and structural information between images [42]. It is used to measure the structural fidelity between the restored image and the reference image. The range of SSIM values is [0, 1]. When SSIM approaches 1, it indicates that the structures of the two images are more consistent and the visual similarity is higher. Its definition is as shown in Equation (Equation 10). Through the above two indicators, the image reconstruction quality and detail restoration ability of each model under different SNR and feature map input conditions can be quantitatively evaluated, providing a reliable basis for model performance analysis.(14)$$\mathrm{PSNR}=10\cdot \log_{10}\left(\frac{{\left(2^{B}-1\right)}^{2}}{\frac{1}{mn}\sum_{i=0}^{m-1}\sum_{j=0}^{n-1}{\left[I\left(i,j\right)-K\left(i,j\right)\right]}^{2}}\right)$$
where the numerator represents the maximum possible pixel value of the image, m and n denote the dimensions of the image, I and K correspond to the denoised image and the original noisy image, respectively, and B stands for the binary representation of pixel values. In this study, the spectra used for evaluation were saved as 8-bit integers, so B is 8.

Figure 13 shows the changes in training and testing loss curves of the proposed de-RDVNet model under the condition of a −10 dB signal-to-noise ratio (SNR) with PNCC feature maps as input. It can be observed that with the progress of training iterations, both the training loss and testing loss of the model continuously decrease and tend to stabilize in the later stage, indicating that the network has good convergence. Among them, the final training set loss converges to 0.405, and the testing set loss converges to 0.402. The two curves are highly consistent in the later stage, indicating that the model has excellent fitting performance and no obvious overfitting phenomenon occurs. Table 1 summarizes the PSNR and SSIM performances of five comparative networks (including de-RDVNet, VDNNet, RIDNet, CBDNet, and DeamNet) under five signal-to-noise ratios (from −10 dB to 0 dB) with three types of feature spectrogram inputs (MFCC, PNCC, and RASTA-PLP). It can be observed from Table 1 that as the SNR increases (i.e., noise interference weakens), the PSNR and SSIM values of each model on the three types of feature maps show a gradual upward trend, which is consistent with the common sense judgment that noise level is positively correlated with reconstruction quality.

To further compare the overall performance of different networks and feature spectrograms across the full SNR range, Table 2 calculates the average PSNR and SSIM values of each network in Table 1 under five SNR conditions. The results show the following: (1) Among all comparative networks, de-RDVNet achieves an average PSNR of 29.8 and SSIM of 0.820 under PNCC feature spectrograms, significantly outperforming other methods, indicating its superior denoising capability in strong noise backgrounds. DeamNet follows next. (2) From the perspective of feature spectrograms, PNCC generally obtains the highest PSNR and SSIM values in all networks. This indicates that PNCC features have the strongest suppression ability for non-stationary noise in this task, with optimal feature robustness and representation capability. In summary, de-RDVNet has obvious advantages in both structural design and feature adaptation, and especially when combined with PNCC spectrograms, it can achieve optimal pest signal denoising and reconstruction performance.

### 4.2. Comparison of Classification Models

Based on the aforementioned denoising experiments, PNCC spectrograms are further selected as feature inputs, and DeamNet, which ranks second in denoising performance, is used as the pre-denoising module. Combinatorial comparisons are conducted with five mainstream lightweight classification models, including ShuffleNet, Swin Transformer, ConvNeXt, MobileViT, and the proposed RDVNet in this paper. These methods cover both convolutional and Transformer architectures, enabling a comprehensive evaluation of the relative performance of RDVNet. We used the same AdamW optimizer, batch size, and loss function. We also used the same input feature map parameters (MFCC, RASTA-PLP, and PNCC). The aim is to verify the recognition performance differences in different classification backbones under the same feature map input and pre-denoising conditions.

As shown in Figure 14, the training loss and testing loss curves of RDVNet during the classification training process are highly consistent and continue to decline with the increase in the number of training epochs, indicating that the model has good convergence and generalization capabilities. After 50 training epochs, the training set loss decreased to 0.198, and the testing set loss was 0.197. The two are almost the same, indicating that the network achieves an optimal fitting state without obvious overfitting. In addition, Figure 11b and Figure 15a show the changing trends of the classification accuracy and F1-score of the five comparative models under five signal-to-noise ratio conditions. It can be observed that as the signal-to-noise ratio decreases, the overall trend of each model still shows a gradual increase in accuracy and F1-score, indicating that the value of the signal-to-noise ratio will affect the recognition ability of the model. Under all signal-to-noise ratio conditions, RDVNet always maintains the highest classification accuracy and F1-score, with the accuracy stabilizing above 90.0%, demonstrating excellent anti-noise recognition capability. Further averaging the classification performance under the five signal-to-noise ratios shows that the average F1-score is 0.878, and the average classification accuracy is 92.8%. In summary, the results show that RDVNet not only performs excellently in denoising tasks, but also outperforms current mainstream lightweight models in classification performance when combined with high-robustness feature map inputs, demonstrating good practicality and promotion value.

### 4.3. Discussion

The comprehensive experimental results show that the integrated denoising and recognition model proposed in this paper exhibits superior performance in multi-signal-to-noise ratio (SNR) environments as follows: (1) In terms of denoising performance, under five SNR conditions from −10 dB to 0 dB, the proposed denoising sub-network de-RDVNet outperforms the four comparative mainstream networks (VDNNet, RID-Net, CBDNet, and DeamNet) in all indicators. Its average peak signal-to-noise ratio (PSNR) is 29.8 and average Structural Similarity (SSIM) is 0.820 under all conditions, ranking first among the five types of models. This fully demonstrates that this method has strong comprehensive capabilities in maintaining image structure and suppressing noise, and the restored image has the highest similarity to the original clean image. (2) In terms of feature spectrogram comparison, among the three types of cepstral spectrograms (MFCC, PNCC, and RASTA-PLP), PNCC shows the highest average PSNR and SSIM in all networks. This indicates that it has stronger suppression ability and feature fidelity for non-stationary background noise and is suitable as a robust feature representation for insect peristaltic sounds. (3) In terms of classification performance: Under the five SNR test conditions, RDVNet’s accuracy and F1-score comprehensively outperform the comparative networks. The classification accuracy under all SNRs exceeds 90.0%, with an average accuracy of 92.8% and an average F1-score of 0.878, verifying its strong classification ability under noisy conditions.

RDVNet achieves optimal results in both denoising and classification tasks, mainly attributed to the following two structural design advantages: (1) The dual-branch structure design of de-RDVNet combines Residual Attention Blocks (RABs) and Hybrid Dilated Residual Attention Blocks (HDRABs), achieving the deep fusion of local and global information through multi-level skip connections. Meanwhile, dilated convolution and downsampling operations jointly expand the receptive field, realizing effective differentiation between insect peristaltic signals and background vibration noise in the feature space, providing a clearer feature foundation for the classification module from the source. (2) The cascaded group attention mechanism in the main classification module explicitly introduces information interaction between multiple heads, avoiding the problem of independent calculation between attention heads in traditional Transformers. By sharing and fusing the contextual information of different attention heads, it improves the ability to capture the overall semantics of features while reducing redundant feature calculations and enhancing efficiency and generalization ability. In addition, the robustness of PNCC feature spectrograms was further verified in the experiment. Due to its use of a nonlinear compression mechanism to suppress non-stationary noise, it has stronger anti-interference ability compared to other feature spectrograms in this research scenario, thereby significantly improving classification accuracy. It should be noted that this paper currently only models the binary classification problem of “with pest/without pest”, mainly focusing on the global distinction between insect peristaltic sounds and external noise vibration sounds in feature frequency bands. Future research can further refine pest categories and perform recognition modeling under more complex classification systems to improve the model’s practicality and versatility in multi-pest detection scenarios. It is worth noting that in extreme noise environments (such as −10 dB), de-RDVNet maintains good image reconstruction quality (PSNR ⩾ 23.0, SSIM ⩾ 0.700) for various feature map inputs, while RDVNet achieves a classification accuracy of 90.0% and an F1-score of 0.810, demonstrating its good anti-noise robustness. Future work will further explore the model’s performance in a wider SNR range and real field scenarios and consider introducing an adaptive SNR discrimination mechanism to enhance model adaptability.

Although this study has achieved remarkable results, there are still the following areas for improvement, and subsequent research will focus on the following directions: (1) Improvement in hardware automatic collection: The current system still relies on manual drilling in the insect hole positioning stage, resulting in limited efficiency for large quantities of timber. In the future, semi-automatic or fully automatic drilling and sensor placement platforms can be developed in combination with mechanical arms or automatic positioning devices to enhance the practical application capability of the system. (2) Lightweight model deployment: The cur-rent model deployment is based on desktop GPU devices, which is not suitable for field environments. In the future, model pruning and inference deployment on embedded platforms such as Jetson and Raspberry Pi can be explored to achieve the portability and real-time performance of pest detection devices. (3) System cloud extension: Currently, the PyQt system only supports local interactive operations. In the future, it can be deployed to cloud platforms through Web front-ends or WeChat mini-programs to achieve remote data upload, recognition inference, and result return, constructing an intelligent cloud-based pest identification platform. In summary, the RDVNet model and integrated system proposed in this paper provide a new idea and tool for the efficient identification of wood-boring pests, with good application prospects. Future research will further deepen and optimize in the directions of fine pest classification, lightweight model deployment, and multi-terminal system interaction.

## 5. Conclusions

This paper proposes a multi-attention integrated model for wood-boring insects identification—the Residual Denoising Vision Network (RDVNet), which effectively realizes the end-to-end optimization of denoising and classification tasks. The experimental results show that RDVNet exhibits excellent robustness and discriminative ability under various noise intensity conditions, significantly outperforming the current four mainstream denoising and classification models. Meanwhile, PNCC feature spectrograms demonstrate more comprehensive information expression ability and stronger anti-interference performance in cepstral feature extraction. Specifically, the main contributions of this paper include the following: (1) High-quality pest signal collection and preprocessing: Based on the LabVIEW software–hardware integration platform, the high-precision collection of original insect peristaltic signals is completed in a vibration-isolated environment. Subsequently, the original signals are segmented, format-converted, and mixed with background conveyor belt noise to generate three types of cepstral feature spectrograms: MFCC, RASTA-PLP, and PNCC. (2) The systematic evaluation of denoising and recognition performance: Comprehensive comparative experiments are conducted with four classic denoising networks (VDNNet, RIDNet, CBDNet, and DeamNet) and four typical classification networks (ShuffleNet, ConvNeXt, Swin Transformer, and MobileViT) under five signal-to-noise ratio conditions from −10 dB to 0 dB. The results show that RDVNet achieves optimal denoising and classification performance. (3) The implementation of the integrated graphical system: To improve visualization analysis capability and user interaction efficiency, a PyQt-based graphical user interface system is developed, realizing the full-process operation of pest vibration signals from collection, processing, feature extraction, and model loading to classification result output. The system has a user-friendly interface, high functional integration, and good practicability and scalability.

## Figures and Tables

**Figure 1 sensors-25-06176-f001:**
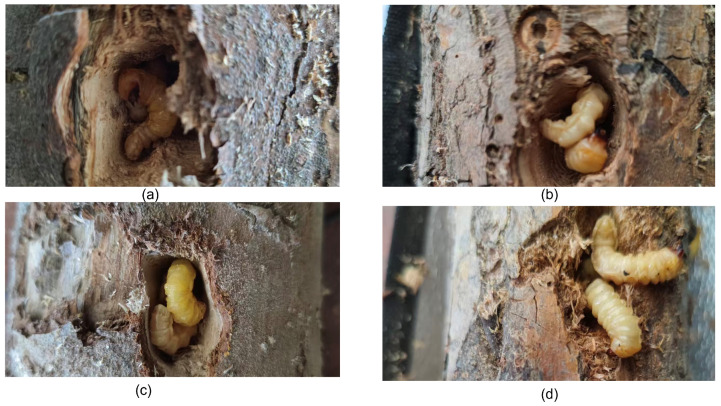
Wood boring insects: (**a**–**d**) show the shapes of boring insects in different wood species.

**Figure 2 sensors-25-06176-f002:**
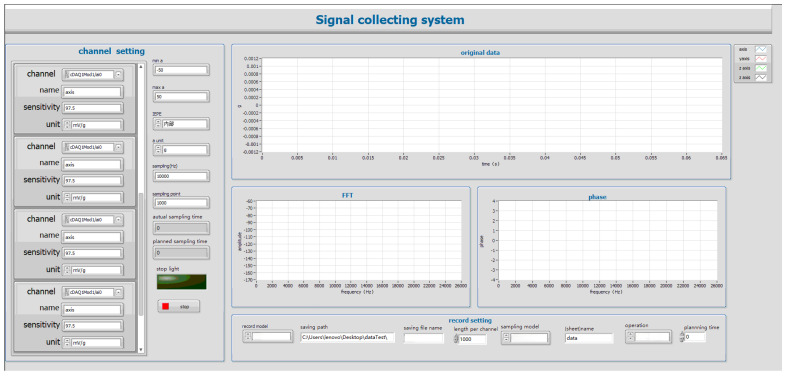
Labview Software System.

**Figure 3 sensors-25-06176-f003:**
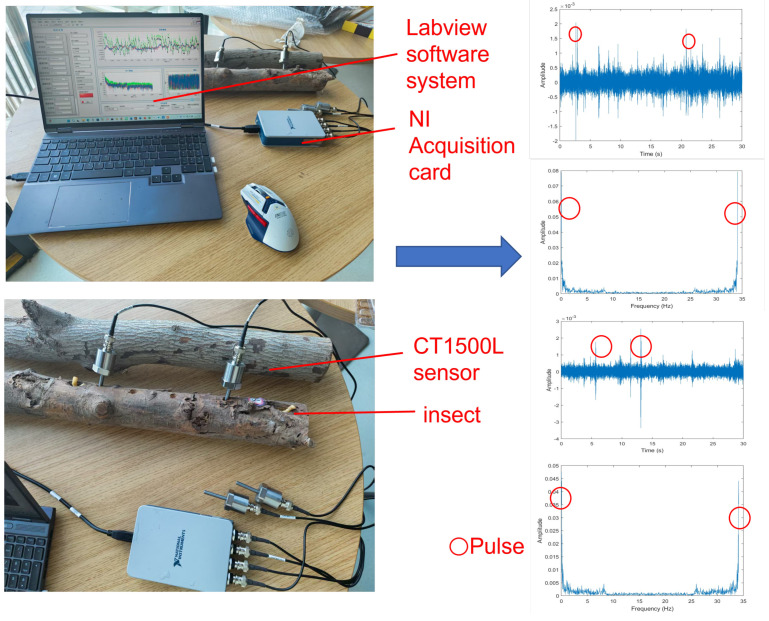
Data collection during the experiment.

**Figure 4 sensors-25-06176-f004:**
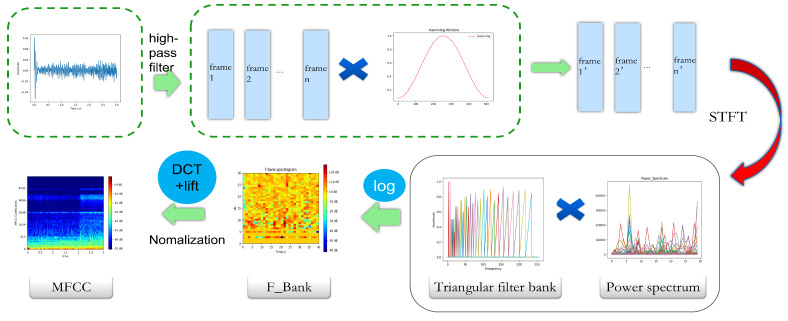
MFCC conversion process.

**Figure 5 sensors-25-06176-f005:**
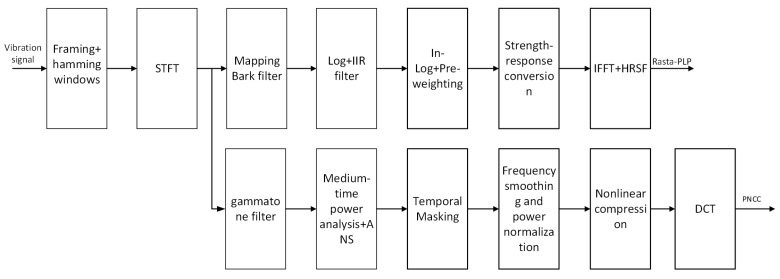
RASTA-PLP, PNCC conversion process.

**Figure 6 sensors-25-06176-f006:**
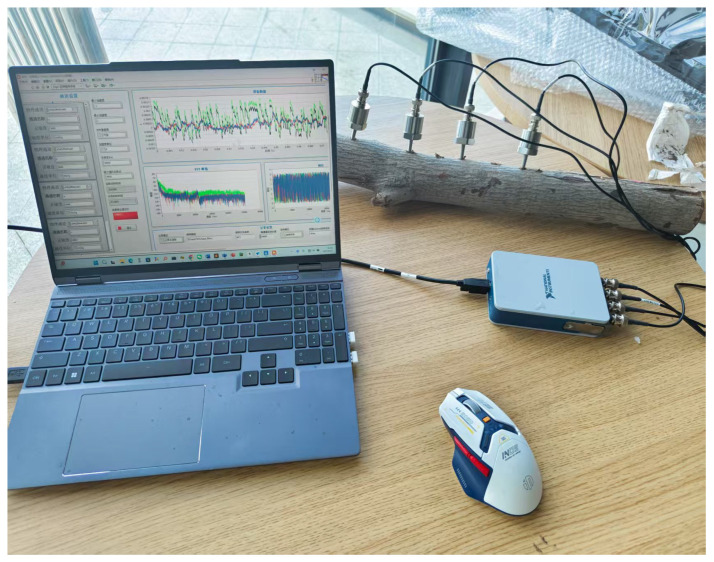
Data collection during actual testing.

**Figure 7 sensors-25-06176-f007:**
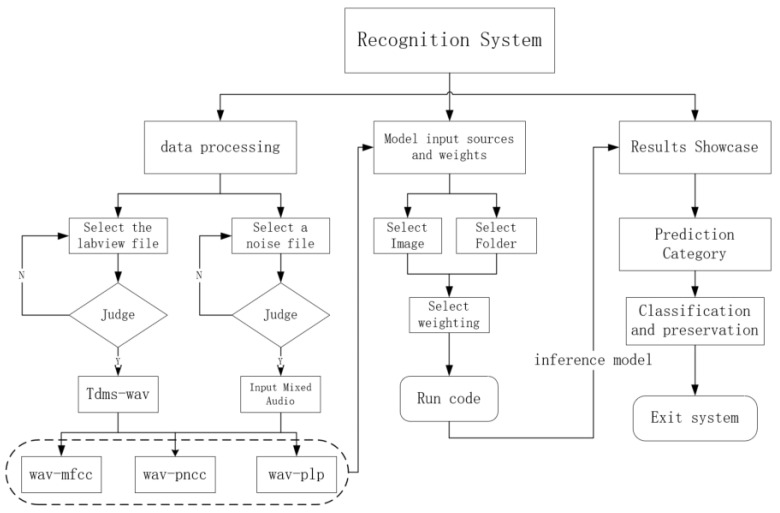
System block diagram of PyQt.

**Figure 8 sensors-25-06176-f008:**
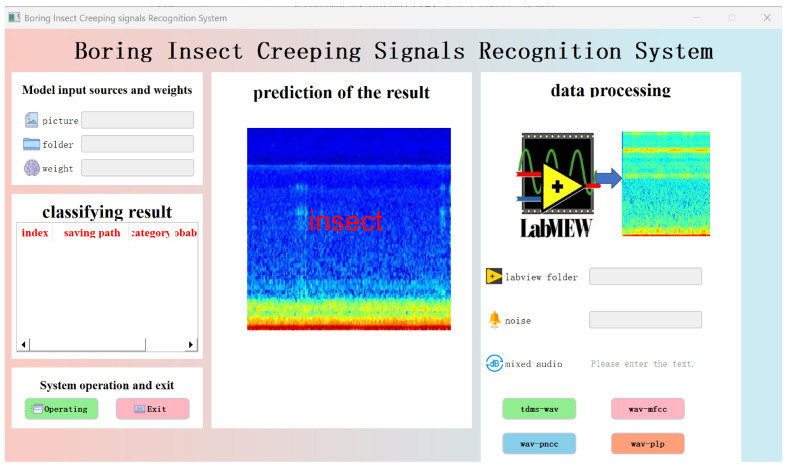
PyQt interface display.

**Figure 9 sensors-25-06176-f009:**
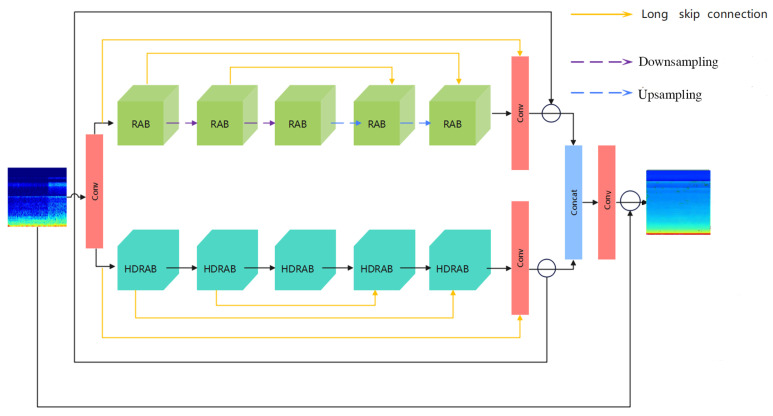
de-RDVNet model structure.

**Figure 10 sensors-25-06176-f010:**
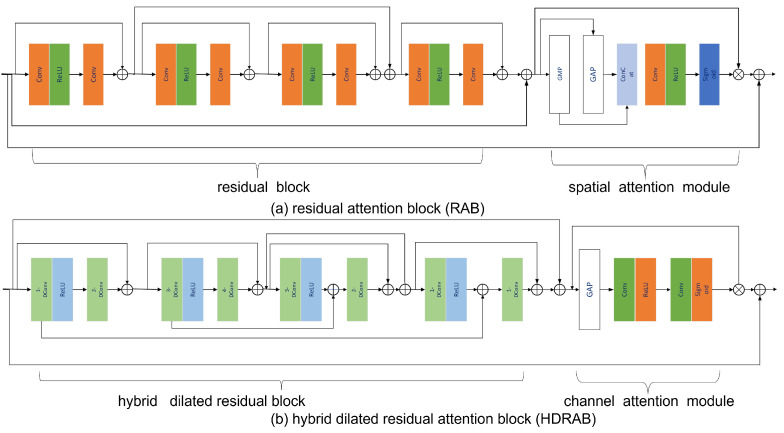
RAB and HDRAB model structure.

**Figure 11 sensors-25-06176-f011:**
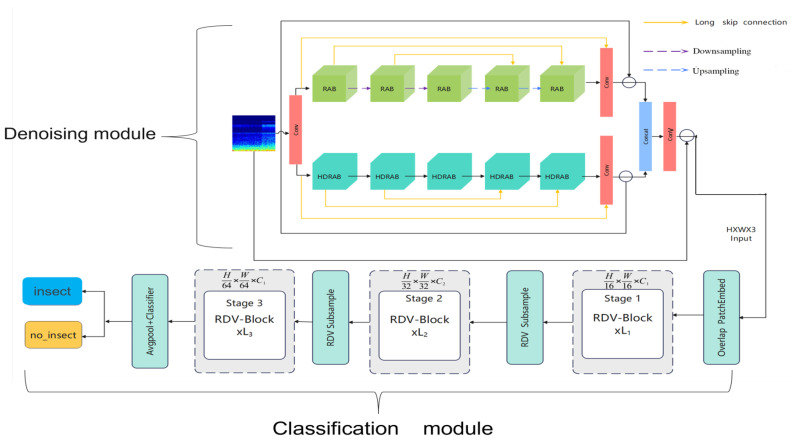
Residual denoising visual network model structure.

**Figure 12 sensors-25-06176-f012:**
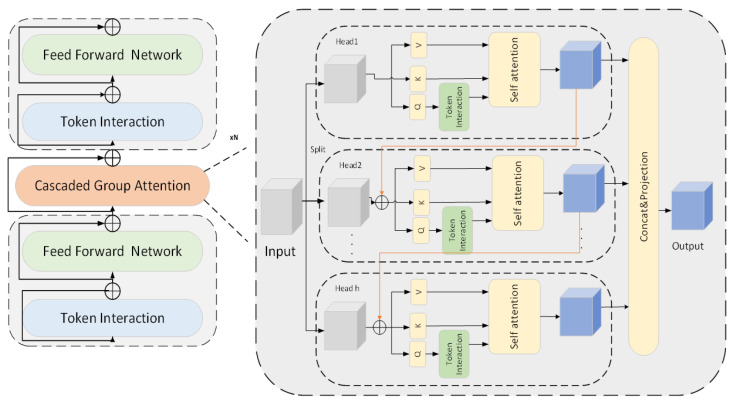
Cascaded group attention model structure.

**Figure 13 sensors-25-06176-f013:**
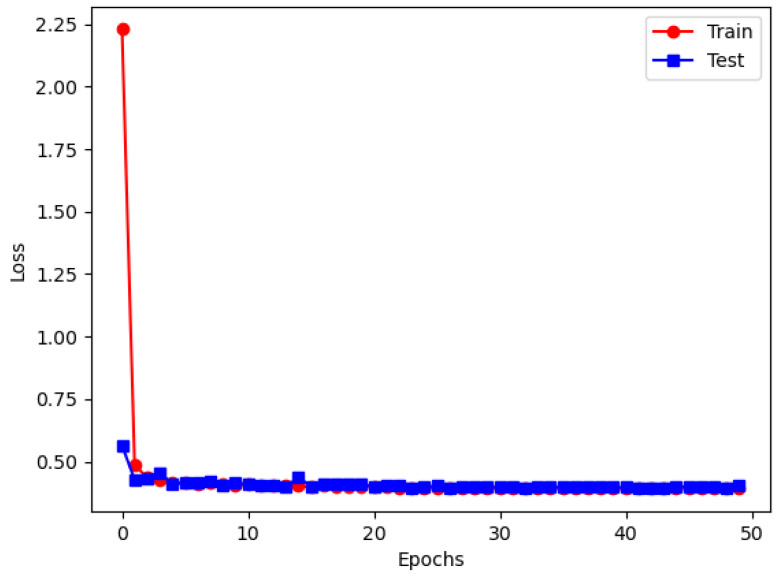
de-RDVNet on the loss image at −10 dB.

**Figure 14 sensors-25-06176-f014:**
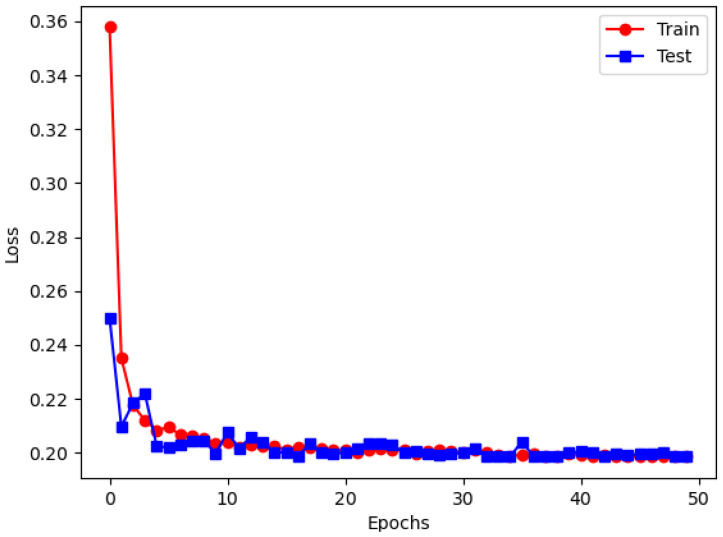
RDVNet in classification loss images at −10 dB.

**Figure 15 sensors-25-06176-f015:**
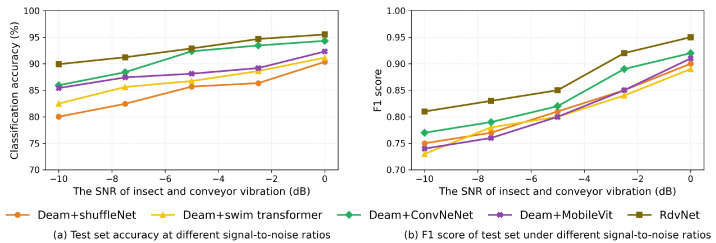
Comparison of accuracy and F1 scores for five models under different signal-to-noise ratios.

**Table 1 sensors-25-06176-t001:** PSNR and SSIM of feature maps for five networks under different SNRs.

Model	Feature Spectrum	−10 dB		−7.5 dB		−5 dB		−2.5 dB		0 dB	
		**PSNR**	**SSIM**	**PSNR**	**SSIM**	**PSNR**	**SSIM**	**PSNR**	**SSIM**	**PSNR**	**SSIM**
CBDNet	MFCC	21.3	0.620	22.5	0.650	23.6	0.670	24.7	0.680	25.3	0.710
	RASTA-PLP	22.2	0.640	23.6	0.660	24.1	0.680	25.7	0.700	26.1	0.720
	PNCC	24.6	0.670	26.6	0.690	25.3	0.710	26.7	0.720	27.1	0.750
RIDNet	MFCC	21.5	0.630	23.7	0.670	24.3	0.690	25.6	0.700	26.7	0.740
	RASTA-PLP	22.2	0.660	24.6	0.690	24.7	0.740	25.8	0.750	27.8	0.760
	PNCC	24.0	0.690	26.8	0.730	25.9	0.760	27.8	0.770	28.8	0.780
VDNNet	MFCC	21.6	0.650	24.8	0.680	25.6	0.720	26.3	0.740	28.2	0.770
	RASTA-PLP	22.7	0.670	25.8	0.700	26.7	0.780	28.0	0.760	29.1	0.800
	PNCC	24.1	0.710	27.6	0.730	27.3	0.800	28.6	0.790	31.5	0.810
DeamNet	MFCC	22.1	0.680	25.0	0.720	26.2	0.740	27.3	0.750	29.9	0.790
	RASTA-PLP	23.4	0.720	26.3	0.740	27.6	0.800	28.9	0.780	31.8	0.820
	PNCC	24.6	0.730	28.1	0.760	28.2	0.810	29.3	0.800	32.1	0.840
de-RDVNet	MFCC	23.6	0.700	25.7	0.750	28.1	0.790	28.1	0.800	30.1	0.820
	RASTA-PLP	24.6	0.730	27.3	0.780	29.3	0.820	29.3	0.840	32.2	0.850
	PNCC	25.7	0.750	29.3	0.780	29.8	0.830	30.1	0.860	33.9	0.880

**Table 2 sensors-25-06176-t002:** Average PSNR and SSIM of feature maps for five networks under different SNRs.

Model	Feature Spectrum	Average PSNR	Average SSIM
CBDNet	MFCC	23.5	0.666
	RASTA-PLP	24.3	0.680
	PNCC	26.1	0.708
RIDNet	MFCC	24.4	0.682
	RASTA-PLP	25.0	0.720
	PNCC	26.7	0.746
VDNNet	MFCC	25.3	0.712
	RASTA-PLP	26.5	0.742
	PNCC	27.9	0.768
DeamNet	MFCC	26.1	0.736
	RASTA-PLP	27.6	0.772
	PNCC	28.5	0.788
de-RDVNet	MFCC	27.1	0.772
	RASTA-PLP	28.6	0.804
	PNCC	29.8	0.820

## Data Availability

The raw/processed data required to reproduce the above findings cannot be shared at this time, as the data also form part of an ongoing study.

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
