# Peer review of "Recognition of Wood-Boring Insect Creeping Signals Based on Residual Denoising Vision Network"

_sensors, 2025, doi:10.3390/s25196176_

Round 1
Reviewer 1 Report
Comments and Suggestions for Authors
Comments and suggestions for Authors
In this manuscript, the authors present a Residual Denoising Vision Network for wood-boring pest detection with noise interference. In general, the study is of great significance for practical applications, and I’m impressed by the solid and detailed work. However, there are several problems existence in this work limit the quality of the manuscript, and need to be deeply revised.
- Since the proposed method is oriented to the practical application of wood-boring pest detection, especially for noise interference environment. Thus, what is the actual detection environment? What kind of background noise interference exists? What are the types of noise? The authors are advised to further clarify the noise interference.
- The literature review deserves a better division. More specifically, the vibration-based detection methods and acoustic-based detection methods are recommended to be summarized separately. The vibration-based method is a contact measurement method using accelerometers, while the acoustic-based method is a non-contact measurement method relying on microphones. Although sound is caused by vibration transmission from objects to air medium, there are obvious differences in measuring instruments, signal physical meanings and processing methods, so the authors should conduct independent method reviews rather than integrated descriptions.
- Although the authors analyze the shortcomings of existing methods, including high model complexity, insufficient robustness and generalization under unstructured and complex background conditions, and lack of unified integrated modeling framework, the summarized main contributions do not fully address the above problems, which makes the existing limitation analysis too general to be meaningful.
- The main contributions of this work are also not summarized well enough. To be specific, the authors claim to propose an integrated framework that can effectively denoising and classification. However, according to the literature review, a similar integration framework has been proposed in reference [18] and has achieved good results. Since the authors did not emphasize the differences and conduct comparative experimental verification, this undoubtedly weakens the novelty of this work. In addition, isn't there any innovative contribution that the authors can highlight in the classification module?
- Are all insert holes in the wood modeled by electric drill and electric saw? Why would authors do that? No real wormholes? The authors should provide detail explanation.
- Why do authors adopt accelerometers to build sound detection models? There are obvious differences in the physical meaning of signals. Using acoustic feature extraction methods to process displacement signals is difficult to understand, and the authors need to explain it clearly and list sufficient reference evidence for explanation. Moreover, I’m wondering why the authors do not use the statistics properties of the displacement signal as features instead of the acoustic features for feature extraction?
- Does the change of sensors placement affect the recognition performance?
- The complete architecture of the model including the denoising module and the classification module and the entire procedure of the training and testing phase need to be supplemented.
- What is the classification loss adopted by the classification module? Cross-entropy? The authors don't mention it. Meanwhile, are the noise reduction loss and the classification loss jointly optimized or independently optimized? The authors are not clear. If optimized independently, the model presents a cascaded architecture rather than an integrated framework. The authors need to explain this in detail. Moreover, how do authors define an end-to-end recognition task? In my opinion, a model that directly operates on the raw waveform signal can be considered as end-to-end recognition in the broad sense. Clearly, the methods proposed by the authors do not fall into this field.
- In the comparison experiment, the introduction of the comparative methods needs to be supplemented to evaluate the fairness of the comparison.
- What is the numerator parameter B in the PSNR formula? According to the standard PSNR public definition, the numerator of the formulation represents the maximum value. Thus, why do the authors choose to employ the maximum pixel value of the spectrogram instead of the more intuitive maximum value of the two-dimensional feature amplitude in the experiment?
- For the developed PyQt software, how are pretrained models deployed? And how does the GPU computing platform interact with LabView locally?
Author Response
Comment 1: Since the proposed method is oriented to the practical application of wood-boring pest detection, especially for noise interference environment. Thus, what is the actual detection environment? What kind of background noise interference exists? What are the types of noise? The authors are advised to further clarify the noise interference.
Response to comment 1:
The actual testing environment was conducted on the workbench at the customs. The main sources of noise are the operating sounds of the surrounding machines, the footsteps of people passing by, and the voices of people, as well as the loud noises from vehicles during their driving. The footsteps of people are transient noise, the voices are non-stationary noise, and the noise from vehicles moving is continuous noise.
The above relevant content has been added in lines 246 to 248 of the text.
Comment 2: The literature review deserves a better division. More specifically, the vibration-based detection methods and acoustic-based detection methods are recommended to be summarized separately. The vibration-based method is a contact measurement method using accelerometers, while the acoustic-based method is a non-contact measurement method relying on microphones. Although sound is caused by vibration transmission from objects to air medium, there are obvious differences in measuring instruments, signal physical meanings and processing methods, so the authors should conduct independent method reviews rather than integrated descriptions.
Response to comment 2:
In terms of detection methods, they are mainly divided into acoustic signal-based detection methods and vibration-based detection methods. In terms of acoustic detection, Chunfeng Dou used an NI acquisition card combined with an acoustic emission sensor SR 150 N to collect acoustic signals, and then used wavelet packets to reconstruct the time-frequency domain signals of pests. The effect of larval number on the number of pulses, duration, and amplitude of the signal was studied. Yufei Bu used the AED-2010L sound detector (built-in sound sensor) combined with the SP-1L probe to collect 4 types of acoustic signals of two types of longhorn larvae, and distinguished them by the amplitude, waveform, pulse and energy of the spectrogram of the time domain map. Senlin Geng used microphones and sound capture cards to collect the sounds of two types of grain storage pests in the soundproof room, and distinguished them by the power spectrum. In terms of vibration signal detection: Piotr Bilski uses a CCLD accelerometer and acquisition card to collect vibration signals and distinguish pest signals from background noise through a support vector machine. Xing Zhang used the SP-1L piezoelectric sensor probe combined with the self-developed vibration sensor to collect vibration signals, and designed TrunkNet to identify pest vibration signals.
The above is on lines 168-181of the text.
Comment3: Although the authors analyze the shortcomings of existing methods, including high model complexity, insufficient robustness and generalization under unstructured and complex background conditions, and lack of unified integrated modeling framework, the summarized main contributions do not fully address the above problems, which makes the existing limitation analysis too general to be meaningful.
Response to comment 3:
The issues I address in lines 184-190 of my paper are: the denoising network's shortcomings in handling non-uniform noise, and the lack of integration between the denoising and classification networks. Comparative denoising experiments addressing non-stationary, transient, and impulsive noise demonstrate that the traditional MFCC method is only moderately robust to non-stationary noise, impulsive noise, and reverberation when extracting spectral information; it is also sensitive to channel convolution mismatch, while PNCC is more robust to non-stationary noise (such as vehicles, footsteps, and mechanical vibrations), sudden interference, and low signal-to-noise ratios (SNRs). My denoising and classification modules are integrated into a single network, as shown in Figure 1 below:
Figure 1 RDVNet
Comment4: The main contributions of this work are also not summarized well enough. To be specific, the authors claim to propose an integrated framework that can effectively denoising and classification. However, according to the literature review, a similar integration framework has been proposed in reference [18] and has achieved good results. Since the authors did not emphasize the differences and conduct comparative experimental verification, this undoubtedly weakens the novelty of this work. In addition, isn't there any innovative contribution that the authors can highlight in the classification module?
Response to comment 4:
The model proposed in Reference 18 is a Dual-Vibration Enhancement Network, a network used for vibration signal enhancement. Four classic classification networks were then used to compare signals that had undergone the vibration enhancement network with those that had not. The two models were not integrated. I apologize for any errors in the description. Because the article is not open source and my limited ability makes it impossible to reproduce, my classification module primarily uses the cross-entropy loss as the loss function.
Comment 5: Are all insert holes in the wood modeled by electric drill and electric saw? Why would authors do that? No real wormholes? The authors should provide detail explanation.
Response to comment 5:
These pests often live inside wood, so inserting the sensor probe into the wood during the experiment required drilling with an electric drill ,as shown in Figure 2 below. By inserting the sensor into the hole, the vibration waves generated by the pests only need to be transmitted in the wood medium, and the noise impact is small. I need to find the pests to further confirm that the amplitude changes in the time-frequency domain were caused by the pests' vibrations, so I had to split the wood with a chainsaw to confirm. However, for actual testing, only drilling with an electric drill was sufficient.
Figure 2 sensor
Figure 3 Data collection during the experiment
When collecting data in the experiment, we only need to use an electric drill to drill a large hole at the wormhole location to find the pest. Then, place two sensors at different distances from the pest to collect data. The other two are placed on the table as a control to observe the changes in the time-frequency domain pulse. The entire environment is in a quiet and low-vibration environment, as shown in Figure 3.
Figure 4 Data collection during actual testing
During the test, four sensors can be arranged at equal distances on the wood to collect signals within each range. as shown in the Figure 4, and the system processes the data into feature maps. The acquisition time is 120s.The system calls the feature maps of each sensor and the weight file to perform inference .Each sensor will output a classified image. The output images are saved in two folders, one is “insect” and the other is “no-insect”. The proportion of the two folders to the total number is counted. Compare the proportion of "insect" folders in each sensor with the set threshold. If the insect ratio of a sensor exceeds the threshold, the wood detected by the sensor is considered to have insects.
Comment 6: Why do authors adopt accelerometers to build sound detection models? There are obvious differences in the physical meaning of signals. Using acoustic feature extraction methods to process displacement signals is difficult to understand, and the authors need to explain it clearly and list sufficient reference evidence for explanation. Moreover, I’m wondering why the authors do not use the statistics properties of the displacement signal as features instead of the acoustic features for feature extraction?
Response to comment 6:
Sound detection methods have drawbacks. The most significant is the extremely low signal-to-noise ratio (SNR) of sound signals. Because the sound of wood-boring pests is so faint, highly sensitive and expensive sound sensors are used. This makes the sound signal very susceptible to external interference during acquisition. Vibration sensors, on the other hand, only require a lack of surrounding vibration. Furthermore, vibration sensors are much cheaper than sound sensors. For more information, refer to the master's thesis "Analysis and Feature Recognition of Vibration Signals from Wood-Boreing Pests." When a vibration sensor is inserted into a hole, the vibration waves generated by the pests only need to propagate through the wood, minimizing noise. However, using sound sensors requires propagation through both the wood and air, making them susceptible to interference from background noise such as wind, human voices, vehicles, and birdsong, drowning out the pest signal. Vibration sensors are more sensitive to weak signals; the amplitude of vibrations generated by pests is typically much smaller than the ambient sound intensity (even less than 1 μm/s²). Being sensitive to low-amplitude, low-frequency mechanical vibrations, they are more likely to detect subtle signs of pests gnawing or crawling.
Both vibration and sound sensors essentially collect "wave signals." Both reflect amplitude changes in the time-frequency domain and can therefore be converted into audio signals. Finally, they are converted into feature maps to extract more features. Currently, research using vibration sensor data to convert it into audio signals is common, such as the following papers:
"Detecting emerald ash borer boring vibrations using an encoder-decoder and improved DenseNet model"
"A time-frequency domain mixed attention-based approach for classifying wood-boring insect feeding vibration signals using a deep learning model"
"Artificial intelligence-based early recognition of feeding sounds of borer pests"
"Lightweight Implementation of the Signal Enhancement Model for Early Wood-Boring Pest Monitoring"
"A CNN-Based Method for Enhancing Boring Vibration with Time-Domain Convolution-Augmented Transformer"
Comment 7: Does the change of sensors placement affect the recognition performance?
Response to comment 7:
Yes, the farther the sensor is from the larvae, the weaker the received signal becomes. When the distance reaches 200mm, the received signal is almost 0.
Comment 8: The complete architecture of the model including the denoising module and the classification module and the entire procedure of the training and testing phase need to be supplemented.
Response to comment 8:
Figure 1 shows the complete structure of the RDVNet model. The upper half does not include the denoising module, while the lower half is the classification module. Because the collected data is subsequently contaminated with noise, the entire framework is used for post-denoising classification. The recognition accuracy and F1 score tested at different signal-to-noise ratios are reported in the paper.
Comment 9:What is the classification loss adopted by the classification module? Cross-entropy? The authors don't mention it. Meanwhile, are the noise reduction loss and the classification loss jointly optimized or independently optimized? The authors are not clear. If optimized independently, the model presents a cascaded architecture rather than an integrated framework. The authors need to explain this in detail. Moreover, how do authors define an end-to-end recognition task? In my opinion, a model that directly operates on the raw waveform signal can be considered as end-to-end recognition in the broad sense. Clearly, the methods proposed by the authors do not fall into this field.
Response to comment 9:
The loss function for the classification module uses a cross-entropy loss, and the loss function for the entire RDVNet model is also a cross-entropy loss. The classification and denoising modules are integrated into RDVNet. The upper half does not include the denoising module, while the lower half is the classification module. Both the denoising and classification modules, as part of RDVNet, share the cross-entropy loss.
The core of the end-to-end task is not "recognizing the original waveform," but rather directly outputting the recognition results from the acquired raw signal, without the need for manual feature engineering. This can be done by directly inputting waveforms or normalized time-frequency images. I extracted feature maps between the waveform and the model, so while it's not end-to-end recognition in the broadest sense, it is a form of end-to-end recognition. This model's recognition of feature maps can compress the original signal into a low-dimensional image with stronger time-frequency correlation, reducing computational effort, easing training difficulty, and facilitating convergence.
Comment 10:In the comparison experiment, the introduction of the comparative methods needs to be supplemented to evaluate the fairness of the comparison.
Response to comment 10:
For the denoising comparison models, we selected four mainstream image denoising networks (VDNNet, RIDNet, CBDNet, and DeamNet). These networks are representative of signal denoising tasks and are often used as benchmarks. For the classification comparison models, we selected lightweight classification networks (ShuffleNet, MobileViT, etc.) and emerging Transformer structures (Swin Transformer, ConvNeXt). These methods cover both convolutional and Transformer architectures, enabling a comprehensive evaluation of the relative performance of RDVNet.
Identical parameter settings: We used the same AdamW optimizer, cosine learning rate schedule, batch size, and loss function. We also used the same input feature map parameters (MFCC, RASTA-PLP, PNCC) to avoid performance imbalances due to hyperparameter differences.
Comment 11:What is the numerator parameter B in the PSNR formula? According to the standard PSNR public definition, the numerator of the formulation represents the maximum value. Thus, why do the authors choose to employ the maximum pixel value of the spectrogram instead of the more intuitive maximum value of the two-dimensional feature amplitude in the experiment?
Response to comment 11:
Using the maximum pixel value of the spectrum graph ensures a fixed numerator as long as the number of bits is determined. Using the maximum amplitude value may result in variations in the amplitude value between data acquisitions. Consequently, comparison experiments cannot be performed under a consistent benchmark.
Comment 12:For the developed PyQt software, how are pretrained models deployed? And how does the GPU computing platform interact with LabView locally?
Response to comment 12:
After each model training, a weight file is generated. You only need to import the weight file into PyQt. LabVIEW is only used for data acquisition. After the data is collected, it is imported into PyQt for data processing and feature map generation.

Reviewer 2 Report
Comments and Suggestions for Authors
The article under review addresses an important issue concerning the detection of wood-destroying insects. This problem is highly relevant not only for environmental protection but also for the field of wood technology. The study presents an attempt to automate the identification of insects based on the vibrations generated by larvae, which constitutes a promising scientific concept. Nevertheless, the work as presented suffers from several serious shortcomings that limit its scientific value and applicability.
First, the use of accelerometers for vibration measurement is, in principle, a sound methodological choice. However, the transducers employed in the study are excessively large. Their installation requires the drilling of relatively wide holes in the wooden samples. This procedure effectively exposes the insects directly, thereby rendering the acoustic investigation redundant. In such a case, the fundamental rationale for employing accelerometric methods is undermined.
Second, the study fails to justify why vibration measurements were not performed without cutting or otherwise damaging the wooden samples. Since vibrations are known to propagate effectively through wood, it would have been entirely feasible to conduct non-invasive experiments. The omission of such an approach raises significant concerns regarding both the design of the experiment and the validity of its results.
Third, the article lacks a sufficient description of how the recorded vibration data were transformed into visual representations. Similarly, the application of the Fast Fourier Transform (FFT) is insufficiently documented. Critical details, such as the choice of window function or the rationale for adopting a 32 kHz sampling frequency, are absent. These omissions hinder the reproducibility and transparency of the presented results, which are fundamental principles of scientific research.
Fourth, the authors do not provide any explanation of how they ensured isolation from extraneous sources of vibration. Environmental vibrations, especially at low frequencies, are notoriously difficult to eliminate in laboratory conditions. This issue is crucial in experiments of this nature, as the presence of uncontrolled background vibrations could compromise the integrity of the measurements.
Taken together, these methodological weaknesses—particularly the reliance on oversized transducers that necessitate destructive intervention in the samples—cast doubt on the extent to which the reported results can genuinely contribute to current methodologies for insect detection in wood. Nonetheless, it should be acknowledged that the general research direction is appropriate and scientifically valuable.
In conclusion, despite addressing an important topic and proposing a potentially promising line of investigation, the article in its current form cannot be recommended for publication. Substantial revisions are required, especially concerning the experimental design, the justification for invasive measurement techniques, and the detailed description of signal processing methods. Only after addressing these issues could the work be considered for dissemination in a scientific journal.
Author Response
Comment 1:
First, the use of accelerometers for vibration measurement is in principle, a sound methodological choice. However, the transducers employed in the study are excessively large. Their installation requires the drilling of relatively wide holes in the wooden samples. This procedure effectively exposes the insects directly, thereby rendering the acoustic investigation redundant. In such a case, the fundamental rationale for employing accelerometric methods is undermined.
Response to comment 1:
These insects often live inside wood, so inserting the sensor probe into the wood during the experiment required drilling with an electric drill. The sensor is shown in Figure 1. I need to find the pests to further confirm that the amplitude changes in the time-frequency domain were caused by the pests' vibrations, so I had to split the wood with a chainsaw to confirm. However, for actual testing, only drilling 6mm hole to allow sensor probe to be inserted with an electric drill was sufficient.
Figure 1 sensor
Figure 2 Data collection during the experiment
When collecting data in the experiment, we only need to use an electric drill to drill a large hole at the wormhole location to find the pest. Then, place two sensors at different distances from the pest to collect data. The other two are placed on the table as a control to observe the changes in the time-frequency domain pulse. The entire environment is in a quiet and low-vibration environment, as shown in Figure 2.
Figure 3 Data collection during actual testing
During the test, four sensors can be arranged at equal distances on the wood to collect signals within each range. as shown in the Figure3, and the system processes the data into feature maps. The acquisition time is 120s.The system calls the feature maps of each sensor and the weight file to perform inference .Each sensor will output a classified image. The output images are saved in two folders, one is “insect” and the other is “no-insect”. The proportion of the two folders to the total number is counted. Compare the proportion of "insect" folders in each sensor with the set threshold. If the insect ratio of a sensor exceeds the threshold, the wood detected by the sensor is considered to have insects.
Comment 2: Second, the study fails to justify why vibration measurements were not performed without cutting or otherwise damaging the wooden samples. Since vibrations are known to propagate effectively through wood, it would have been entirely feasible to conduct non-invasive experiments. The omission of such an approach raises significant concerns regarding both the design of the experiment and the validity of its results.
Response to comment 2:
When the vibration sensor is inserted into the hole, the vibration waves generated by the pests only need to be transmitted in the wood medium, and the noise impact is small. It can be closer to the location where the pests are active (worm tunnels, around larvae), and the received vibration amplitude is larger. Due to the shortened propagation distance and reduced attenuation, the signal waveform is clearer and the characteristic pulse is more obvious. However, when the sensor is attached to the wood, the signal needs to pass through the wood medium and the air medium, resulting in signal attenuation and loss of frequency components. In terms of noise resistance: wood acts as a barrier to effectively isolate airborne environmental noise (human voices, wind noise, vehicle noise). The received signal is mainly the structural vibration inside the wood, and the signal-to-noise ratio is significantly improved. In addition, in the actual detection process: only some 6mm large holes need to be drilled in the wood, which is not too destructive to the wood. The following documents all use invasive detection:
"Detecting emerald ash borer boring vibrations using an encoder-decoder and improved DenseNet model"
"A time-frequency domain mixed attention-based approach for classifying wood-boring insect feeding vibration signals using a deep learning model"
"Artificial intelligence-based early recognition of feeding sounds of borer pests"
"Lightweight Implementation of the Signal Enhancement Model for Early Wood-Boring Pest Monitoring"
"A CNN-Based Method for Enhancing Boring Vibration with Time-Domain Convolution-Augmented Transformer"
Comment 3: Third, the article lacks a sufficient description of how the recorded vibration data were transformed into visual representations. Similarly, the application of the Fast Fourier Transform (FFT) is insufficiently documented. Critical details, such as the choice of window function or the rationale for adopting a 32 kHz sampling frequency, are absent. These omissions hinder the reproducibility and transparency of the presented results, which are fundamental principles of scientific research.
Response to comment 3:
As mentioned in the Data Acquisition and Collection section of the article, after acquiring TDMS format files using LabVIEW, they are converted into audio signals and segmented. They are then converted into MFC, PLP, and PNCC spectra. This conversion process can be implemented using Python internal library code. A Fast Fourier Transform length of 1024 preserves the characteristics of short-duration insect pulses while maintaining spectral smoothness. A Hamming window is used as the window function, offering a good compromise between suppressing sidelobe leakage and maintaining mainlobe resolution, making it suitable for processing non-stationary insect vibration signals. A sampling rate of 32 kHz was chosen in the experiment based on the following considerations: the main effective frequencies of vibration signals generated by wood pests, such as crawling and gnawing, are concentrated in the 0–10 kHz range (refer to relevant literature on pest vibration monitoring). According to the Nyquist sampling theorem, the sampling rate should be at least twice the highest frequency component of the signal. Therefore, choosing 32 kHz ensures that the high-frequency characteristics of insect vibrations are captured without distortion while also balancing storage and computational efficiency. An example of MFC image conversion is shown in Figure 4:
Figure 4 MFCC conversion process
Comment 4: Fourth, the authors do not provide any explanation of how they ensured isolation from extraneous sources of vibration. Environmental vibrations, especially at low frequencies, are notoriously difficult to eliminate in laboratory conditions. This issue is crucial in experiments of this nature, as the presence of uncontrolled background vibrations could compromise the integrity of the measurements.
Response to comment 4:
All data collection experiments were conducted in a relatively quiet, low-vibration indoor environment, away from large machinery and vehicle traffic to minimize external vibration sources. Anti-seismic facilities are installed at the bottom of the test bench, which attenuated low-frequency mechanical vibrations transmitted from the ground. In each experiment, in addition to inserting sensor probes into wood with insect holes, we also placed control sensors on the surface of wood without insect holes to simultaneously record background vibration signals. The vibration sound of pests is a high-frequency signal in a quiet environment. In subsequent data processing, we compared the signals from the insect hole sensors with those from the control sensors to eliminate irrelevant signals introduced by external vibrations, thereby improving data reliability.

Round 2
Reviewer 1 Report
Comments and Suggestions for Authors
There are still some issues that require further clarification by the authors.
- Since the authors claim that noise comes from a variety of sound sources in the surrounding environment and has non-stationary characteristics. So how do the authors control the signal-to-noise ratio in the experiment to a fixed value (-10dB, -7.5dB, -5dB, -2.5dB) rather than a dynamic range? This is obviously not in line with the practical applications. Furthermore, considering that vibration sensor is employed in this work, the sound pressure variations generated by the environmental noise mentioned by the authors are scarcely likely to exert an influence on the accelerometers. Meanwhile, according to the additional information provided by the authors in their reply, if the signal received by the sensor is almost zero when the distance from the larva is 200mm, then for most noises with wavelengths far greater than 200mm, the impact on the collected signal is almost zero as well. Therefore, the proposed denoising methods seemingly deviate from the actual application scenarios. This is a very serious issue that demands careful attention from the authors.
- The authors need to further explain the scientificity and rationality of using acoustic feature analysis methods such as MFCC, PNCC, and RASTA-PLP to extract vibration features collected by accelerometers. Specifically, MFCC, PNCC, and RASTA-PLP capture sound pressure changes by simulating the nonlinear perception characteristics of the human ear and are accompanied by voice parameter settings, which have completely different physical meanings from the displacement changes caused by vibration. Moreover, among the supplementary references provided by the authors, four are about the combination of raw vibration signals and deep learning models, and one is about the integration of electroacoustic signals and deep learning methods. However, there is no prior case of directly applying acoustic feature analysis methods to vibration signals, which cannot provide support for this work. Therefore, the authors are required to provide well-substantiated justifications for their selected feature analysis methods in order to adequately establish their validity and ensure the persuasiveness of the study to the readers.
- The MFCC, PNCC, RASTA-PLP are typical speech feature extraction methods, which belong to obvious feature engineering rather than end-to-end methods in speech-related tasks. Therefore, it is suggested that the author refer to the original sources to provide the clearest description that does not mislead readers.
- What is the specific value of the parameter B in the PSNR formula? Please provide it directly in the text. Additionally, since B is set based on pixels, will the PSNR value change if different color gamut spaces are used to present the spectrogram? In the face of this potential issue, it seems that the maximum value of the normalized amplitude spectrum is the best solution for providing a consistent benchmark.
Author Response
Comment 1: Since the authors claim that noise comes from a variety of sound sources in the surrounding environment and has non-stationary characteristics. So how do the authors control the signal-to-noise ratio in the experiment to a fixed value (-10dB, -7.5dB, -5dB, -2.5dB) rather than a dynamic range? This is obviously not in line with the practical applications. Furthermore, considering that vibration sensor is employed in this work, the sound pressure variations generated by the environmental noise mentioned by the authors are scarcely likely to exert an influence on the accelerometers. Meanwhile, according to the additional information provided by the authors in their reply, if the signal received by the sensor is almost zero when the distance from the larva is 200mm, then for most noises with wavelengths far greater than 200mm, the impact on the collected signal is almost zero as well. Therefore, the proposed denoising methods seemingly deviate from the actual application scenarios. This is a very serious issue that demands careful attention from the authors.
Response to comment 1:
The signal-to-noise ratio (SNR) was kept constant to test the model's noise-resistance. The denoising network utilizes a multi-attention mechanism, which prioritizes and retains the vibration signals of borer pests during the denoising process. If the entire network (RDVNet) performs well at a constant SNR, then accurate recognition will be achieved across a dynamic range exceeding this SNR. This primarily tests the model's generalization capabilities. During the noise addition process, we also collected real-world noise, and each signal segment was paired with an inconsistent noise segment to closely simulate a real-world environment. In field testing, we achieved an 80% accuracy rate for insect vibration signals collected under significant noise.
The SNR was kept constant, which is a common practice in the field of vibration enhancement: first, real-world vibration noise was collected, then precisely mixed with the insect "net signal" at a target SNR. This "factor control" approach allowed for a horizontal comparison of the robustness of different models and features. (In this article, the dataset and evaluation set were constructed at five SNR levels.)
For details, refer to Tables 1 and 2 in "Lightweight Model Design and Compression of CRN for Trunk Borers' Vibration Signals Enhancement" for the model's vibration enhancement performance at five signal-to-noise ratios. Figure 9. Comparison of the accuracy of the signals after each pruning and enhancement model and then entering the two classification models.
https://doi.org/10.3390/f14102001
Tables 2 and 3 in "Lightweight Implementation of the Signal Enhancement Model for Early Wood-Boring Pest Monitoring" show the vibration enhancement performance of the model at five signal-to-noise ratios for their respective datasets. https://doi.org/10.3390/f15111903
Section 2.3 of "Dataset Construction" in "A CNN-Based Method for Enhancing Boring Vibration with Time-Domain Convolution-Augmented Transformer" mentions random mixed noise and borer vibration sounds at signal-to-noise ratios of -10, -7.5, -5, -2.5, and 0dB. Table 3 compares the vibration enhancement effects of the model at these five signal-to-noise ratios. Figure 6. Comparison of the classification accuracy of the signals at these five signal-to-noise ratios after various enhancement models and then input into VGG16.
https://doi.org/10.3390/insects14070631
In the article "Detecting Emerald Ash Borer Boring Vibrations Using an Encoder-Decoder and Improved DenseNet Model," 2.1.2 Dataset Construction mentions random mixed noise and emerald ash borer vibrations at signal-to-noise ratios of -10, -7.5, -5, -2.5, and 0 dB. Table 5 and Figure 16 show a comparison of the precision and recall rates of various models at these five signal-to-noise ratios.
https://doi.org/10.1002/ps.8442
In "Enhancement of Boring Vibrations Based on Cascaded Dual-Domain Features Extraction for Insect Pest Agrilus planipennis Monitoring," 2.1.2. Data Construction mentions the use of random mixed noise and borer vibration sounds at a signal-to-noise ratio of -10dB.
https://doi.org/10.3390/f14050902
In "A Waveform Mapping-Based Approach for Enhancement of Trunk Borers’ Vibration Signals Using Deep Learning Model," 2.1.2. Dataset Production mentions the use of random mixed noise and borer vibration sounds at a signal-to-noise ratio of -10dB. https://doi.org/10.3390/insects13070596
“Multi-Channel Time-Domain Boring-Vibration-Enhancement Method Using RNN Networks” Figure 11 shows a comparison of the accuracy of two classification models after the signals have been subjected to various enhancement models at five signal-to-noise ratios.
https://doi.org/10.3390/insects14100817
The 200 mm wavelength conclusion addresses the directional propagation and attenuation of the insect-wood system. Environmental vibration often forms multiple input points through the ground, container deck, and wood, triggering acceleration responses at the sensor installation location. Its spatial path differs from that of the insect source. The 200 mm wavelength judgment primarily applies to the far-field aeroacoustic measurements. Accelerometers measure elastic and bending waves in solids and the global and local modal responses of structural components. Local accelerations, determined by phase and boundary coupling conditions, can be significant and are not directly constrained by the "wavelength of airborne sound." Vibrations from surrounding machinery and vehicle traffic can also affect the signal.
Comment 2: The authors need to further explain the scientificity and rationality of using acoustic feature analysis methods such as MFCC, PNCC, and RASTA-PLP to extract vibration features collected by accelerometers. Specifically, MFCC, PNCC, and RASTA-PLP capture sound pressure changes by simulating the nonlinear perception characteristics of the human ear and are accompanied by voice parameter settings, which have completely different physical meanings from the displacement changes caused by vibration. Moreover, among the supplementary references provided by the authors, four are about the combination of raw vibration signals and deep learning models, and one is about the integration of electroacoustic signals and deep learning methods. However, there is no prior case of directly applying acoustic feature analysis methods to vibration signals, which cannot provide support for this work. Therefore, the authors are required to provide well-substantiated justifications for their selected feature analysis methods in order to adequately establish their validity and ensure the persuasiveness of the study to the readers.
Response to comment 2:
Our selection of MFCC, PNCC, and RASTA-PLP is not motivated by the assumption that vibration signals are equivalent to sound pressure. Instead, we consider these three types of cepstral features to be a universal representation framework: a filter bank, nonlinear mapping, and spectral envelope decorrelation. This framework can be applied to any scalar one-dimensional time series signal (including acceleration). In our system, the larval activity signals and ambient noise collected by sensors are uniformly converted into cepstral coefficient spectra (MFCC/PNCC/RASTA-PLP). The gnawing/rubbing behavior of trunk-boring larvae exhibits a repetitive-impact-resonant energy distribution accompanied by a slowly fluctuating activity rhythm. Cepstral coefficients are naturally adept at characterizing envelopes and resonance peaks and are more stable under noise interference. We do not treat "human ear physiology" as a physical assumption, but rather consider these "perception-inspired" steps as a mature engineering approach to spectral feature construction.
MFCCs for vibration: In the paper "Improved Mel-Frequency Cepstral Coefficients for Compressors and Pumps Fault Diagnosis with Deep Learning Models," Cabrera et al. proposed a two-dimensional MFCC representation for pump/compressor vibration and multi-fault classification. The abstract proposes a two-dimensional representation model for vibration signals, using Mel-Frequency Cepstral Coefficients (MFCCs) and their first two derivatives as features. Figure 4 illustrates the use of an accelerometer.
https://doi.org/10.3390/app14051710
Piezoelectric vibration sensors measure displacement, acceleration, and velocity based on their physical properties.
Section 2.1.1 Data Collection and Screening in the paper "Detecting Emerald Ash Borer Vibrations Using an Encoder-Decoder and Improved DenseNet Model" describes the use of piezoelectric sensors to collect vibrations, measuring structural vibrations rather than airborne sound pressure. Figure 6 demonstrates the use of MFCC images as model input.
https://doi.org/10.1002/ps.8442
Page 7 of "A Time-Frequency Domain Mixed Attention-Based Approach for Classifying Wood-Boring Insect Feeding Vibration Signals Using a Deep Learning Model" describes data acquisition using a piezoelectric vibration sensor. Figure 10 shows the use of MFCC images as model input.
https://doi.org/10.3390/insects15040282
“Deep Learning Model Compression for Real-Time Detection of Wood-Boring Insect Vibration”: Data acquisition and processing in this paper uses a piezoelectric vibration sensor. Figure 4 shows the use of MFCC images as model input.
10.12171/j.1000−1522.20200100
Similarly, the PNCC and PLP algorithms, as feature spectrum extraction methods, can also be applied to extract signals from wood-boring insects.
Comment 3:
The MFCC, PNCC, RASTA-PLP are typical speech feature extraction methods, which belong to obvious feature engineering rather than end-to-end methods in speech-related tasks. Therefore, it is suggested that the author refer to the original sources to provide the clearest description that does not mislead readers.
Response to comment 3:
We agree with the reviewer's point: MFCC, PNCC, and RASTA-PLP are all engineered features typical of speech, rather than end-to-end methods. In this study, they serve only as a front-end for extracting cepstral coefficient feature maps, mapping the one-dimensional vibration sequence output by the accelerometer into a two-dimensional feature map, which is then fed as input to the subsequent network. The relevant content has been revised in the original paper.
Comment 4:
What is the specific value of the parameter B in the PSNR formula? Please provide it directly in the text. Additionally, since B is set based on pixels, will the PSNR value change if different color gamut spaces are used to present the spectrogram? In the face of this potential issue, it seems that the maximum value of the normalized amplitude spectrum is the best solution for providing a consistent benchmark.
Response to comment 4:
In this study, the spectrograms used for evaluation were stored as 8-bit integer quantization, resulting in a value of B = 8, corresponding to a peak value of 255. To prevent chromaticity from interfering with objective metrics, we uniformly converted the color spectrograms to YCbCr space and calculated PSNR only on the Y (luminance) component. PSNR calculations are based on the values in the pixel array, not the display device's "presentation gamut/gamma." In our unified processing pipeline, the input spectrogram is first read as an sRGB array, then a fixed YCbCr conversion is applied, with only the Y component used in the calculation. Therefore, the display's color gamut/gamma does not affect the values used in our calculations, and PSNR remains unchanged with changes in the "presentation gamut." To avoid issues such as scale error masking, sensitivity to abnormal peaks, and cross-sample incomparability caused by normalizing each image to , we do not use the "maximum normalization caliber" as the primary evaluation metric. This will "wash out" amplitude scale errors, become sensitive to abnormal peaks/rare pulses, and cause a flattening of the global dynamic range. This can lead to poor comparability across samples and scenes, misalignment with commonly used image objective metric toolchains, and mask engineering issues related to "energy conservation."
